# The cure rate after different treatments for mucosal leishmaniasis in the Americas: A systematic review

**Janaína de Pina Carvalho** [1]*, **Sarah Nascimento Silva**[1], **Mariana Lourenço Freire**[1], **Líndicy Leidicy Alves**[1], **Carolina Senra Alves de Souza**[1,2], **Gláucia Cota**[1]

**1** Pesquisa Clínica e Políticas Públicas em Doenças Infecto-Parasitárias, Fundação Oswaldo Cruz, Belo Horizonte, Minas Gerais, Brazil, **2** Coordenação Estadual de Laboratórios e Pesquisa em Vigilância da Subsecretaria de Vigilância em Saúde da Secretaria do Estado da Saúde de Minas Gerais, Belo Horizonte, Minas Gerais, Brazil

* janaina.carvalho@fiocruz.br

**Data Availability Statement:** All relevant data are within the manuscript and its Supporting Information files.

## Abstract

### Background

Mucosal or mucocutaneous leishmaniasis is the most severe form of tegumentary leishmaniasis due to its destructive character and potential damage to respiratory and digestive tracts. The current treatment recommendations are based on low or very low-quality evidence, and pentavalent antimonial derivatives remain strongly recommended. The aim of this review was to update the evidence and estimate the cure rate and safety profile of the therapeutic options available for mucosal leishmaniasis (ML) in the Americas.

### Methodology

A systematic review was conducted in four different databases and by different reviewers, independently, to evaluate the therapeutic efficacy and toxicity associated with different treatments for ML. All original studies reporting cure rates in more than 10 patients from American regions were included, without restriction of design, language, or publication date. The risk of bias was assessed by two reviewers, using different tools according to the study design. The pooled cure rate based on the latest cure assessment reported in the original studies was calculated grouping all study arms addressing the same intervention. The protocol for this review was registered at the International Prospective Register of Systematic Reviews, PROSPERO: CRD42019130708.

### Principal findings

Twenty-seven original studies from four databases fulfilled the selection criteria. A total of 1,666 patients with ML were treated predominantly with pentavalent antimonials in Brazil. Other interventions, such as pentamidine, miltefosine, imidazoles, aminosidine sulfate, deoxycholate and lipidic formulations of amphotericin B (liposomal, lipid complex, colloidal dispersion), in addition to combinations with pentoxifylline, allopurinol or sulfa were also considered. In general, at least one domain with a high risk of bias was identified in the included

**Funding:** GC is funded by National Counsel Technological and Scientific Developments – CNPq (grant number: 301384/2019). The funders had no role in study design, data collection and analysis, decision to publish, or preparation of the manuscript.

**Competing interests:** The authors have declared that no competing interests exist.

studies, suggesting low methodological quality. The pooled cure rate based on the latest cure assessment reported in the original studies was calculated grouping all study arms addressing the same intervention. It was confirmed that antimony is still the most used treatment for ML, with only moderate efficacy (possibly increased by combining with pentoxifylline). There is already evidence for the use of miltefosine for ML, with a cure rate similar to antimony, as observed in the only direct meta-analysis including 57 patients (OR: 1.2; 0.43–3.49, $I^2 = 0$). It was possible to gather all descriptions available about adverse events reported during ML treatment, and the toxicity reflected the pattern informed in the manufacturers' technical information.

## Conclusions

This study provides an overview of the clinical experience in the Americas related to ML treatment and points out interventions and possible combinations that are eligible to be explored in future well-designed studies.

## Author summary

Mucosal leishmaniasis (ML) is a deforming leishmaniasis clinical form related to functional damage and stigmatization. This disease is caused mainly by *L. braziliensis* and predominates in neglected populations in the Americas, where approximately 2000 cases occur per year. There are few ML clinical trials, which makes the current treatment supported by fragile scientific evidence. In this study, we carried out an extensive literature search to gather the accumulated evidence for ML treatment. Twenty-seven studies with different designs were included with a total of 1,666 patients with ML treated. The results confirmed that antimony is still the main drug used for ML treatment, with only a moderate cure rate, an efficacy possibly increased by pentoxifylline combination. Miltefosine was found to be an alternative, with a cure rate similar to antimonials standard doses. In turn, the high toxicity of amphotericin B deoxycholate was clearly demonstrated, generating low cure rates due to early interruption of treatment. Other alternatives such as pentamidine, imidazoles, and aminosidine, were evaluated in a small number of cases and presented variable cure rates. The quality of the studies is low, and there is great heterogeneity in the definitions of cure, which limits a global analysis of the data. More and well-designed studies are needed to guide ML treatment recommendations.

## Introduction

Mucosal (ML) or mucocutaneous (MCL) leishmaniasis is considered the most severe form of tegumentary leishmaniasis (TL) due to its destructive character and potential for functional loss in the respiratory and digestive tracts. Mucous involvement characteristically occurs years after the appearance of skin lesions, although it can also be detected simultaneously with or without apparent cutaneous involvement [1,2]. Although the species *L. panamensis*, *L. guyanensis* and *L.amazonensis* [3,4] have also been implicated with mucosal involvement, in addition to *L. infantum* and *L. major* [5], according to the World Health Organization, the term 'mucocutaneous leishmaniasis' should be applied only to the New World disease, caused by *L. braziliensis* [6].

Relative to the magnitude of the problem, approximately half of the cases of the American TL cases are registered in Brazil [7], which has a robust national epidemiological surveillance system based on the mandatory notification for all forms of leishmaniasis. Based on the Notifiable Diseases Information System (SINAN) of the Ministry of Health, between 2007 and 2020, 288,133 cases of TL were registered in Brazil, and in 17,079 (6%) ML was the clinical presentation indicated in the notification form [8]. In the Americas, the Pan American Health Organization has made efforts to gather notification data from all countries through the SisLeish platform. According to this database, between 3.4% and 4.3% of ML or MCL cases have been observed in relation to the total number of CL cases reported in recent years, estimates likely underreported [7].

The established therapeutic options available for ML require parenteral administration and have a high toxicity profile [6]. According to the treatment guidelines for the Americas, the current recommendations are based on low or very low-quality evidence and the pentavalent antimonial (Sb$^v$) derivatives (with or without oral pentoxifylline) remain strongly recommended, given the scarcity of studies evaluating other alternatives [9].

Other systematic reviews have been published on the subject [1,10,11], but they are outdated or have included only the few randomized controlled studies (RCTs) available. Assuming that the available evidence in RCTs is insufficient to support treatment recommendations, we decided to carry out a more flexible search and analyze all the accumulated experience in ML treatment. Therefore, the aim of this review was to update the evidence and estimate the cure rate with the therapeutic options available for LM in the Americas based on a comprehensive and critical literature search. The safety profile related to each of the therapeutic options was also evaluated, when reported, and considered the secondary outcome of this review.

## Methods

### Protocol and registration

The protocol was registered at the International Prospective Register of Systematic Reviews (PROSPERO) under protocol CRD42019130708. This review was conducted following the Preferred Reporting Items for Systematic Reviews and Meta-Analyses (PRISMA) [12] statement (S1 Table).

### Eligibility criteria

The eligibility criteria were established according to a guiding question defined based on the acronym PICO: (P) Population: subjects with leishmaniasis acquired in the Americas and mucosal involvement; (I) Intervention: any pharmaceutical drug treatment; (C) Comparator, if applicable: any other therapy, placebo, or no treatment; (O) Outcome: cure rate. Secondary outcomes: adverse events and relapse rates. RCTs, nonrandomized clinical trials, and observational studies (cohorts or case reports) were considered eligible.

### Exclusion criteria

The exclusion criteria were nonoriginal studies (literature review, letters, replies, editorials, guidelines, publications presenting the same patients previously described in another article); in vitro studies, or those addressing nonhuman participants or specific populations (immunosuppressed or failed patients) and studies in which, for each intervention evaluated, fewer than 10 patients were treated. Studies that did not present the number of ML patients cured, when there were CL patients in the sample, were also excluded. There were no language or publication date restrictions.

### Information sources and search strategies

The literature search was concluded on December 15th, 2021, and the sources were MEDLINE (PubMed), Embase, LILACS, and Web of Science. Supporting information (**S2 Table**) describes in detail the search strategy for each database and the number of studies retrieved. A manual search was also performed in the references included in the previously published systematic reviews.

### Study selection and data extraction

The titles and abstracts of the retrieved articles were independently evaluated by two reviewers (JPC and CSAS) using Rayyan software [13]. Disagreements were resolved by consensus or by a third reviewer (MLF, GC). After the initial selection, four authors (JPC, SNS, MLF, and LLA) performed a new screening in pairs of the selected full texts to confirm eligibility or identify exclusion criteria.

Four review authors (JPC, SNS, MLF and LLA) independently extracted data from the included studies using a standard data extraction form to collect characteristics of the studies, population, outcomes, and adverse events. The data extracted were then confirmed by a second reviewer (JPC, SNS and GC). If more than one publication reported the same patients, the study with the largest number of patients was selected and the other studies were excluded to avoid duplicate cases. When the same patients were described in more than one publication but for the presentation of different outcomes (e.g., cure and relapse rates), they were considered, but patients were counted only once.

### Outcomes

The outcome of interest was cure rate at any moment after the end of treatment, using the intention-to-treat approach, regardless of how the author described it in the original publication and expressed as the number of cases cured by the total number of cases treated with a given intervention. Thus, the losses observed during the follow up were considered therapeutic failures. In order to maintain consistency among the definitions of cure used in the studies, the cure criterion assumed in the analyzes was as strict as possible, that is, even for authors who considered a re-epithelialization above 90% as cure, if the data on complete healing were available, these were the ones adopted as cure. Considering D1 as the first day of treatment, and using the definitions proposed by Olliaro (2013) [14], we arbitrarily adopted the following intervals to standardize the cure rate data extraction at different times after treatment: cure assessment performed between D60 and D135 was assumed to be D90; cure assessment performed between D136 and D270 was assumed to be D180; and cure assessment performed between D271 and D390 was assumed to be D360. For relapse rate estimation, only patients considered cured previously were considered. The safety of antileishmanial therapy was captured in each study as the number of adverse events (AE) per total patients (or treatments, alternatively, in case it is the only information available) evaluated. The rates were expressed as percentages and presented together with their respective 95% confidence intervals (CIs) calculated using the Mantel Haenszel random effects model. The adopted AE classification originally used (severity, intensity or other) was also obtained and, when available, was described for each study.

### Data synthesis and statistical analysis

Comprehensive Meta-Analysis software v.3.0 was used to perform a one-group meta-analysis of study arms using a given treatment (pooled rates) based on the latest cure assessment

reported in the original studies. Clinical cure rates were calculated according to the intention-to-treat analysis: the analysis was based on the total number of randomly assigned participants, irrespective of how the original study's authors analysed the data. These unadjusted indirect comparisons were compared with direct comparisons, when available. We used the inconsistency ($I^2$) statistic to evaluate heterogeneity. If significant heterogeneity was found, the results from the random effects model were emphasized and summary measures were analyzed as limited information, looking for differences in studies. A random effects model is a strategy that allows the interstudy heterogeneity to be incorporated through a broad CI, generating a more conservative estimate of the measure of the effect. For a global analysis within a given therapy with subgroups, a mixed effects analysis was used: a random effects model was used to combine studies within each subgroup and a fixed effect model was used to combine subgroups and yield the overall effect. The study-to-study variance (tau-squared) was assumed to be the same for all subgroups; this value was computed within subgroups and then pooled.

The Medical Dictionary for Regulatory Activities terminology (MedDRA), version 25.0, an international medical terminology developed under the auspices of the International Council for Harmonization of Technical Requirements for Pharmaceuticals for Human Use (ICH) [15], was used to standardize the safety data in this review. In brief, each AE reported in the primary study was recorded and classified in accordance with the Preferred Term (PT), High Level Group Terms (HLGL), and System Organ Classes (SOC). In the same hierarchical order of the stratification by system-organ, the MedDRA system includes the term "investigations" to designate alterations of complementary diagnostic tests referring to several systems.

## Quality of evidence assessment

Two pairs of independent researchers (MLF/SNS and MLF/LLA) assessed the risk of bias using specific tools according to the study design. RCTs were evaluated using the Cochrane risk of bias score (RoB 2) [16,17] and nonrandomized clinical trials and prospective cohort studies were evaluated using the Newcastle Ottawa Scale (NOS) [18]. In addition, the modified NOS (Murad, 2018 [19]) was used to assess bias in retrospective observational studies (mostly case series), as previously performed by others [20–24]. RoB 2 is based on five domains: (1) randomization process; (2) deviations from intended interventions; (3) missing outcome data; (4) measurement of the outcome; and (5) selection of the reported result. In turn, the NOS tool includes nine items that can be categorized into three dimensions: 1) selection of study groups, 2) comparability of groups, and 3) determination of the results of interest. The adopted tool for case report assessment is based on four domains (selection, ascertainment, causality and report) and eight items. This tool was adapted, as suggested by Murad (2018), resulting in five items. Three of these five items, related to report and outcome aspects, received double weight. To standardize the risk of bias assessment, it was defined that the minimum follow-up time of 90 days [14] and a loss to follow-up of less than 20% would be the parameters for a low risk of bias study. For all domains, the risk was assumed to be high if there was not enough information to assess the quality.

## Results

### Studies and population characteristics

The search identified 1,104 records from the Embase (350), MEDLINE (323), Web of Science (231) and LILACS (200) databases. The process of study selection and reasons for exclusion are summarized in a PRISMA flow diagram (**Fig 1**). After removal of duplicates, 822 records had their titles and abstracts evaluated and 679 were excluded. From the 143 selected studies, 142 were read in full to confirm their eligibility and to extract the data. After numerous

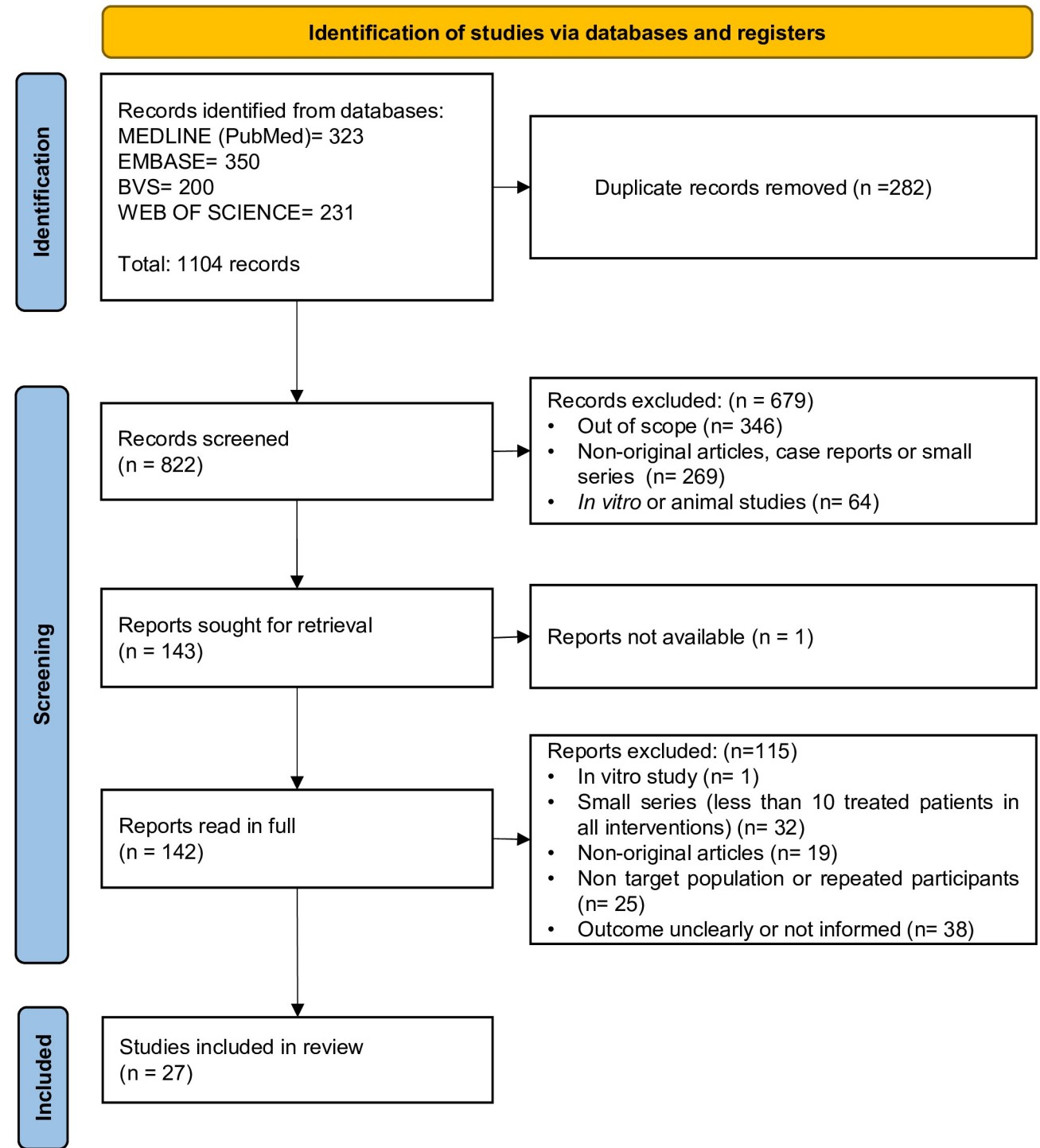

**Fig 1. PRISMA flow diagram of the study selection process.**

attempts, we were unable to obtain Fernandes's (1990) [25] study for full text evaluation. After 115 exclusions, 27 articles were included for quantitative synthesis.

Among the 27 studies included, 7 were RCTs, 15 were observational studies (12 retrospective and 3 prospective) and 5 were nonrandomized experimental studies. The main

methodological characteristics of the studies, namely, design, definitions of ML case and cure, follow-up time and origin are presented in **Table 1**. Most studies were conducted in Brazil (17 studies), and the other countries were Bolivia (3), Peru (4), Argentina (1), Panama (1) and Ecuador (1). In total, 1,666 patients with ML were gathered, and the interventions evaluated were pentavalent antimonials [26–29], both meglumine antimonate (MA) [29–40] and sodium stibogluconate (SSG) [29,41–45]; different lipid formulations of amphotericin, such as liposomal preparation (L-AMB), a lipid complex (ABLC) and a colloidal dispersion (c-AMB) [28,31,33,46,47] and deoxycholate amphotericin B (d-AMB) [28,33,48,49]; aminosidine sulfate (AS) [50,51]; pentamidine (PENT) [28,33]; miltefosine (MF) [30,32,48,52]; imidazoles (IMIDZ) [26,28,31,33,53]; and few combined therapies based on antimony derivatives and sulfa [26], allopurinol [42] or pentoxifylline [34,40].

All studies included symptomatic patients, that is, individuals with clinical manifestation of mucosal involvement, and most of them with confirmation of *Leishmania* infection either by direct examination, culture, polymerase chain reaction (PCR) or presence of amastigote in the histological examination. Some studies also considered a Montenegro skin test (MST), serological tests, epidemiological exposition, and a therapeutic test, as alternative confirmatory criteria.

Most studies defined cure as complete epithelialization of all lesions, associated with the disappearance of inflammatory signs (infiltration, oedema, redness). For some authors, based on the assessment of a lesion severity score, a clinical improvement above 90% was considered a cure [32,48,52]. For these studies, the total number of patients with complete healing was also reported, allowing us to extract this as the study cure outcome, to maintain consistency with the other studies. In one study, the definition adopted for cure was not informed [26]. Some series described more than one course of treatment for the same patient who did not achieve cure after the first treatment attempt. In these cases, the outcome considered was the number of patients cured with the first course of a given intervention.

In general, the authors used the terms "relapse" or "recurrence" indistinctly. In 11 studies, the condition was defined as the reappearance of the activity of an already healed mucosa [27,28,31,33,36,41,42,45,46,50,51]. For three studies, relapse could occur after incomplete healing [42,45,50], and for the other five studies, relapse could be new lesion onset [27,28,31,33,45]. Time for cure assessment and follow-up varied widely among studies and the initial date from which the follow-up accounting starts was not a consensus among the authors, some counting from the first and others from the last day of treatment. Most studies presented cure assessment within one year from the end of treatment but at different moments. The follow-up time in general was also variable, ranging from one month [40] to 10 years [36]. When the follow-up time was not previously set but the patients' follow-up average was presented, this information was considered (**Table 1**).

As shown in **Table 2**, the population gathered in this review was mostly made up of adults, with an average age varying from 30.7 to 74 years. Only five studies [26,27,33,38,49] included children, and in four studies, the authors did not report the patient's ages [34,36,37,48]. Based on studies providing the sample size stratified by sex, the male:female ratio was 1077:437, as expected due to the recognized predominance of men affected by ML. The nasal mucosa was the most affected site, and the average duration of symptoms before treatment varied from 3 to 272 months.

ML severity was classified using different criteria across the studies. The Llanos-Cuentas (1997) classification [42], which combines mucosal lesion extension and severity of symptoms, was used in five studies [31,33,39,42,50]. In turn, Soto's (2007) [48] classification was adopted in three studies [32,48,52] and Lessa's classification (2012) [54] was used in two others [30,40]. Several other criteria were used for classifying the disease severity, including the presence of

**Table 1. Characteristics of the included studies.**

| Year, Author [reference] | Study design | Country (cases) | Treatment (patients) | ML case definition | Cure definition | Relapse definition | Cure assessment | Follow-up (months) |
|---|---|---|---|---|---|---|---|---|
| 2019, Sampaio [30] | RCT | Brazil (38) | MA 20 mg/kg/day, EV, for 30 days. (18) <br><br> MF, 50mg (1.3 a 2 mg/kg/day), Oral, for 28 days. (20) | Active mucosal lesion and epidemiological exposition in addition to Leishmania visualization (culture, direct examination, histology), or at least two of the followings: compatible histology, positive MST or positive IIFR. | Complete epithelialization and the absence of inflammation signs within four years after the end of treatment | NR | At 90 days, 180 days and 4 years after the end of treatment. | 48 |
| 2019, Santos [28] | Observational retrospective | Brazil (105) | ABLC 1–4 mg/kg/dia. (13), limit to 2500mg <br><br> d-AMB 1 mg/kg/dia. (14), limit to 2500mg <br><br> L-AMB (1–4 mg/kg/day), limit to 2500mg (32) <br><br> Itraconazole 200mg/day, Oral, for 6 weeks. (10) <br><br> Pentamidine 4 mg/kg/day. Oral, for 10 days. (11) <br><br> Sb$^v$ 20 mg/kg/day, EV, for 30 days (25) | PCR, culture, or Leishmania visualization during histological examination or immunohistochemistry. | Complete epithelialization within 1 year after the end of the treatment | Resurgence after complete healing or a new lesion for one year follow up | At 1 year after the end of the therapy | 12 |
| 2018, Cataldo [38] | Observational retrospective | Brazil (27) | MA 5mg/kg/day, IM or EV, for 30 to 120 days. (27) | Active mucosal lesion and visualization of parasites in culture | Complete epithelialization and disappearance of inflammatory signs and no relapse recurrence. | NR | During treatment, 1, 3, 6, 9, 12, 18, 24, 36, 48 and 60 months after the end of treatment. | 60 |
| 2018, Pedras [31] | Observational retrospective | Brazil (35) | L-AMB 3mg/kg/day, mean 2,550 mg; range: 2.10–3.00 mg, EV, for 14 days; range: 11–20 days. (9) <br><br> Fluconazole 600mg, range: 450-900mg, Oral, for 120 days; range: 49–396 days. (9) <br><br> MA 20mg/kg/day, EV, for 30 days. (17) | Active mucosal lesion and one or more of the following: positive PCR or a compatible histological exam associated with a positive therapeutic test and a positive immunological test (MST or IIFR) | Complete epithelialization | Resurgence after complete healing or a new lesion at any moment. | At 6 months after the end of the treatment. | 12 |
| 2017, Cincurá [40] | Observational retrospective | Brazil (251) | MA 20mg/kg/day, EV with/without pentoxifylline, 1,200 mg/day, Oral, for 30 days. (251) | Active mucosal lesion and a positive PCR or a compatible histology associated with a positive MST | Complete epithelialization and no relapse at 90 days after initiation of therapy. | NR | At 90 days after the start of treatment. | 3 |

(*Continued*)

**Table 1.** (*Continued*)

| Year, Author [reference] | Study design | Country (cases) | Treatment (patients) | ML case definition | Cure definition | Relapse definition | Cure assessment | Follow-up (months) |
|---|---|---|---|---|---|---|---|---|
| 2015, Cunha [46] | Observational retrospective | Brazil (29) | L-AMB mean total cumulative dose 32.5 mg/kg; range: 18.2–55.2 mg/kg, EV. (29) | Active mucosal lesion and at least one of following: epidemiological exposition, visualization of parasites in smear, culture or histology, positive PCR or a positive immunological test (MST or serology) | Complete re-epithelialization 6 months after the end of the treatment. | Resurgence after complete healing during a one year of follow up | At 6 months after the end of the treatment. | 6 |
| 2014, Bustos [32] | RCT | Argentina (19) | MA 10 a 20 mg/kg/day; limit to 850 mg/day; for 28 to 35 days. (10) | Active mucosal lesion and visualization of parasites in culture or a positive PCR or a positive MST (not clearly reported) | Improvement > 90% in relation to the initial score at 12 months after the end of treatment. | NR | At 2, 6, 9 and 12 months after treatment. | 12 |
| | | | MF 2.5 a 3.3 mg/kg/day, Oral, for 28 to 35 days. (9) | | | | | |
| 2014, Rocio [47] | Observational retrospective | Brazil (16) | L-AMB 3–5 mg/kg/day; upper daily limit of 200 mg; EV, until 40 mg/kg. (16) | Active mucosal lesion and visualization of parasites (smear or culture or histology) or a positive immunological test (MST, ELISA, IIFR) or | Complete healing within one year after the end of treatment | NR | At 12 months after the end of treatment | 12 |
| 2009, Amato [33] | Observational retrospective | Brazil (140) | c-AMB 3 mg/kg/day; upper daily limit of 200 mg; EV, limit to 40 mg/kg. (9) | Active mucosal lesion and visualization of parasites in smear or immunohistochemistry or a compatible histology associated with a positive therapeutic test. | Complete healing within 1 year after the end of the treatment | Resurgence after complete healing or a new lesion during a 1 year of follow-up | 1 year after the end of the therapy. | at least 18 |
| | | | d-AMB 1 mg/kg/day (upper daily limit of 50 mg), EV, limit to 1500 mg. (17) | | | | | |
| | | | L-AMB 3–5 mg/kg/day (upper daily limit of 200 mg), EV, limit to 40 mg/kg. (4) | | | | | |
| | | | Itraconazole 200 mg/day, Oral, for six weeks. (15) | | | | | |
| | | | MA 20mg/kg/day, EV, for 30 days. (73) | | | | | |
| | | | Pentamidine 4 mg/kg/day, EV, limit to 2000 mg. (22) | | | | | |
| 2009, Soto [52] | Observational prospective | Bolivia (21) | MF, 50mg; 2.5 to 3.3 mg/kg/day, Oral, for 6 weeks. (21) | Active mucosal lesion and visualization of parasites in smear or culture or a positive MST and a suggestive CL scar | Improvement of > 90% in relation to the initial score | NR | 1 year after the end of treatment (Group C) | 12 |

(*Continued*)

**Table 1.** (Continued)

| Year, Author [reference] | Study design | Country (cases) | Treatment (patients) | ML case definition | Cure definition | Relapse definition | Cure assessment | Follow-up (months) |
|---|---|---|---|---|---|---|---|---|
| 2007, Llanos-Cuentas [50] | RCT | Peru (38) | AS, 14 mg/kg, IM, for 21 days. (21) / MA 20mg/kg/day, EV, for 28 days. (17) | Active mucosal lesion and visualization of parasites by culture, histology, and/or PCR on a biopsy specimen. | Complete epithelialization and disappearance of inflammatory signs at 1 year after the end of treatment. | Resurgence after complete or incomplete healing | At the end of treatment and every 3 months for 1 year. | 12 |
| 2007, Machado [34] | RCT | Brazil (23) | MA 20mg/kg/day, EV, for 30 days. (12) / MA 20mg/kg/day, EV, plus pentoxifylline, 400 mg, Oral, 3 times daily, for 30 days. (11). | Active mucosal lesion and by at least 2 of the following laboratory methods: a positive MST, visualization of parasites in culture or compatible histology | Complete epithelization and disappearance of inflammatory signs at 150 days after initiation of therapy. | NR | At 150 days after initiation of therapy. | 24 |
| 2007, Soto [48] | Observational prospective | Bolivia (97) | d-AMB 1mg/kg/day, EV, for 45 days. (19) / MF, 50mg, 2.5 a 3.3 mg/kg/day, Oral, for 28 days. (78) | Active mucosal lesion and visualization of parasites in smear or culture or a positive MST associated with a skin scar suggestive of CL | Improvement of > 90% in relation to the initial score | NR | At the end of therapy, and at 2, 6, 9, and 12 months after the end of therapy | 12 |
| 2006, Bermudez [41] | Observational prospective | Bolivia (18) | SSG, 20 mg/kg/day, for 20 days. (18) | Active mucosal lesion and positive MST or a positive parasitological test (culture, smear or PCR). | Complete scarring (for ulcers) or flattening (nodular lesions) associated with absence of infiltration and disappearance of inflammatory signs within 3 months of the end of treatment, and no relapse within 12 months of the end of treatment. | Resurgence after complete healing for one year follow up | At 12 months after the end of treatment | 12 |
| 2005, Name [27] | Observational retrospective | Brazil (148) | Sb$^v$ 20mg/kg/day for 30 days. (148) | Active mucosal lesion and epidemiological exposition and a simultaneously associated skin lesion suggestive of CL or a positive parasitological test (culture, smear or inoculation in hamster), immunological (IIRI and MST) or compatible histology | Epithelialization, absence of local infiltration and erythema within one year after the end of treatment | Resurgence after complete healing or a new lesion for one year follow up | At the end of treatment and within one year after the end of treatment | 12 |
| 2004, Calvopina [53] | non-randomized experimental | Ecuador (13) | Itraconazole, 400 mg/day in two daily doses, Oral, for 12 weeks. (13) | Active mucosal lesion and positive a parasitological test, (smear, cultures, PCR) or a positive MST | Complete epithelialization (for ulcers) or complete resolution (for non-ulcers) | NR | At 3, 6 e 12 months after treatment | 12 |

(*Continued*)

**Table 1.** (*Continued*)

| Year, Author [reference] | Study design | Country (cases) | Treatment (patients) | ML case definition | Cure definition | Relapse definition | Cure assessment | Follow-up (months) |
|---|---|---|---|---|---|---|---|---|
| 2000, Oliveira-Neto [39] | non-randomized experimental | Brazil (31) | MA 5mg/kg/day, IM, for 30 days. (21)<br><br>MA 5mg/kg/day, IM, for 45 days. (10) | Active mucosal lesion (mild to moderate) and at least four of the following five criteria:<br>a) compatible clinical lesions and disease history.<br>b) a positive MST.<br>c) suggestive histology.<br>d) negative tests for other diseases that may affect the oronasal mucous membranes notably leprosy, paracoccidioidomycosis and syphilis.<br>e) demonstration of leishmania in cultures, histology and/or PCR. | Complete epithelialization (for ulcers) or complete resolution (for non-ulcers) associated with disappearance of inflammatory signs | NR | At the end of treatment and at 3,6,12 months. | 12 |
| 1997, Llanos-Cuentas [42] | RCT | Peru (81) | SSG 20mg/Kg/day, EV, for 28 days. (41)<br><br>SSG 20mg/Kg/day, EV, for 28 days plus allopurinol 20 mg/Kg/day, Oral for 28 days. (40) | Active mucosal lesion (severe or moderate) and demonstration of leishmania in culture and/or PCR | Complete epithelialization (for ulcers) or complete resolution (for non-ulcers) associated with disappearance of inflammatory signs | Resurgence after complete or incomplete healing or a new lesion | At 12 months after treatment | 12 |
| 1996, Romero [51] | non-randomized experimental | Brazil (13) | AS 16mg/kg/day, IM, for 20 days. (13) | Active mucosal lesion and compatible histology or positive inoculation in hamster or positive MST or positive IIFR. | Complete remission and no relapse during one-year | Resurgence after complete healing for one year follow up | At 12 months after treatment | Mean 12,6 (11,5–14) |
| 1995, Oliveira [36] | Observational retrospective | Brazil (51) | MA 10 a 20mg/kg/day, for 30 days. (51) | Active mucosal lesion and demonstration of leishmania in culture, histology or a positive inoculation in hamster | Remission of lesions for a minimum period of 1 year | Resurgence after complete healing for one year follow up | At 12 months after treatment | Mean 124 |
| 1994, Franke [43] | RCT | Peru (40) | SSG 20mg/kg/day EV for 28 days. (20)<br><br>SSG 20mg/kg/day EV for 40 days. (20) | Active mucosal lesion and demonstration of leishmania in culture | Complete epithelialization (for ulcers) or complete resolution (for non-ulcers) associated with disappearance of inflammatory signs | NR | At 12 months after treatment | 12 |
| 1993, Zocoli [26] | Observational retrospective | Brazil (303) | Ketoconazole 3 tablets/day for 21 days; 2 tablets/day for 28 days and 1 tablet/day for 90 days, Oral. (25)<br><br>Sb$^v$ 25 ampoules, IM, on alternate days. (98)<br><br>Sb$^v$ 25 ampoules plus Sulfa, IM, on alternate days. (180) | Two or more diagnostic tests: compatible histology, positive MST, demonstration of leishmania in direct exam or a positive therapeutic test. | NR | NR | At 6 months after treatment | at least 6 |

(*Continued*)

**Table 1.** (Continued)

| Year, Author [reference] | Study design | Country (cases) | Treatment (patients) | ML case definition | Cure definition | Relapse definition | Cure assessment | Follow-up (months) |
|---|---|---|---|---|---|---|---|---|
| 1991, Kopke [37] | RCT | Brazil (17) | MA 14mg/kg/día, EV, 3 series of 30 days with an interval of 15 days. (10)<br><br>MA 28mg/kg/día, EV, 3 series of 30 days with an interval of 15 days. (7) | Active mucosal lesion associated with epidemiological exposition, positive MST or IIFR, or compatible histology | Complete healing | NR | At two years after treatment | 24 |
| 1991, Saenz [44] | non-randomized experimental | Panamá (16) | SSG 20 mg/kg/day, EV, for 28 days. (16) | Active mucosal lesion associated with demonstration of leishmania in culture c or positive MST. | Complete epithelialization (for ulcers) or complete resolution (for non-ulcers) and no relapse during follow-up. | NR | At 12 months after treatment | 12 |
| 1990, Franke [45] | non-randomized experimental | Peru (29) | SSG 20 mg/kg/day, EV, for 28 days. (29) | Active mucosal lesion and demonstration of leishmania in culture | Complete epithelialization (for ulcers) or complete resolution (for non-ulcers) associated with disappearance of inflammatory signs | Resurgence after complete or incomplete healing or a new lesion during 12 months of follow-up. | At the end of treatment and 3 months after the end of treatment. | 12 |
| 1989, Sampaio [29] | Observational retrospective | Brazil (56) | MA 28mg/kg/day, EV, 10–12 days, 3 series with an interval of 15 days. (26)<br><br>Sbᵛ–MA or SSG– 20mg/kg/day, EV, 14 to 85 days; mean 30 days. (12)<br><br>SSG 10mg/kg/day, EV, 30 days. (18) | Active mucosal lesion, a positive therapeutic test and at least one of the following three conditions: MST, positive, parasitological evidence or compatible histology. | Complete healing within one month after the end of treatment and no relapse in to 69 months after treatment. | NR | Within 1 month after the end of treatment | 30<br><br>8<br><br>12 |
| 1960, Sampaio [49] | Observational retrospective | Brazil (11) | d-AMB, 1mg/kg to 1.25mg/kg, EV, on alternate days (total dose 725–1,850 mg). | Active mucosal lesion associated with positive MST, or compatible histology | Complete healing | NR | NR | Variable: few weeks to one year. |

ABLC: Amphotericin B, lipid complex. AS: Aminosidine Sulfate. CL: cutaneous leishmaniasis. c-AMB: Amphotericin B colloidal dispersion. d-AMB: deoxycholate amphotericin B. ELISA (Enzyme Linked Immuno Sorbent Assay). EV: Intravenous. IIFR: Indirect Immunofluorescence Reaction. IM: intramuscular. L- AMB: Liposomal amphotericin B. MA: Meglumine antimoniate. ML: mucosal leishmaniasis. MF: Miltefosine. MST: Montenegro Skin Test. NR: No reported. PCR: Polymerase Chain Reaction. RCT: Randomized Clinical Trial. Sbᵛ: Pentavalent antimony. SSG: Sodium stibogluconate.

extranasal involvement [38,45], presence of septum perforation [34,44,53] and number of lesions [51]. No severity classification was presented in 11 studies [26–29,36,37,41,43,46,47,49].

Nine studies presented *Leishmania* species identification in a variable percentage of patients [30,32,38,42–45,48,53]. Among them, *L. braziliensis* was the most frequently reported species. *L. panamensis* and *L. amazonensis* were identified from three [44] and one patient [32], respectively.

**Table 2. Characteristics of the population with mucosal leishmaniasis.**

| Year, Author [reference] | Treatment (N) | Age ($\bar{x}$ or $M_d$ ± SD [Min-Max]) years (n) | Gender (male: female) | Duration of symptoms before therapy (months ± SD) [Min-Max] (n) | Mucosal lesion site: n/N | ML clinical classification: n/N | Previous CL: n/N | Leishmania species: n/N |
|---|---|---|---|---|---|---|---|---|
| 2019, Sampaio [30] | MA (18) | $\bar{x}$: 50.8 ± 13 (18) | 9:9 | 141.5 ± 152.5 | Only nasal: 35/38 (92.1%) Only oral: 1/38 (2.6%) Nasal + pharynx: 2/38 (5.3%) | The majority of the patients had **moderate** to **severe** disease and were at least stage IV according to the clinical classification proposed by Lessa et al, 2012 [54]. | NR | *L. braziliensis*: 22/38 (58%) |
| | MF (20) | $\bar{x}$: 61.2 ± 11.3 (20) | 9:11 | 112.4 ± 133.3 | | | | |
| 2019, Santos [28] | ABLC (13) d-AMB (14) L-AMB (32) Itraconazole (10) PENT (11) Sb$^v$ (25) | $\bar{x}$: 59.1 ± 14.2 (71) | 43:28 | <12 (12) 12–60 (16) 60–120 (30) > 120 (13) | Nasal: 55/71 (77%) Palate: 14/71 (20%) Pharynx: 18/71 (25%) Larynx: 10/71 (14%) | NR | 37/71 (52%) | NR |
| 2018, Cataldo [38] | MA-LD, RJ (12) | $M_d$: 52 [12–75] (12) | 12:0 | $M_d$: 3 [1–24] | Nasal exclusive: 3/12 (25%) | Less severe (nasal exclusive): 3/12 Severe (other than nasal mucosa): 9/12 | NR | *Leishmania* (*V.*) *braziliensis*: 12/12 (100%) |
| | MA-LD, OS (15) | $M_d$: 52 [12–80] (15) | 10:5 | $M_d$: 24 [1–704] | Nasal exclusive: 10/15 (67%) | Less severe (nasal exclusive): 10/15 Severe (other than nasal mucosa): 5/15 | | *Leishmania* (*V.*) *braziliensis*: 6/15 (40%) |
| 2018, Pedras [31] | Fluconazole (9) | $M_d$: 74 [51–85] (9) | 6:3 | $M_d$: 24 [3–240] | Nasal: 33/35 (94%), with 30/33 presenting septum ulcer or perforation. Oral cavity/mouth: 4/35 (11%) Pharynx: 4/35 (11%) | According to Llanos-Cuentas et al, 1997 [42] Mild: 9/9; moderate: 0/9; severe: 0/9 | NR | NR |
| | L-AMB (9) | $M_d$: 68 [63–80] (9) | 7:2 | $M_d$: 36 [2–420] | | Mild: 6/9; moderate: 2/9; severe: 1/9 | | |
| | MA (17) | $M_d$: 39 [16–64] (17) | 11:6 | $M_d$: 12 [6–168] | | Mild: 14/17; moderate: 1/17; severe: 2/17 | | |
| 2017, Cincurá [40] | MA or MA + pentoxifylline (251) | $M_d$: 38.5 (327)* | 220:107* | $M_d$ (1995–2004): 6 $M_d$ (2005–2014): 3 | Nasal cavity: 318/326 (98%) Pharynx: 36/326 (11%) Oral cavity: 18/326 (6%) Larynx: 5/326 (1.5%) | According to Lessa, 2012 [54]: Stage I: 32/312 Stage II: 100/312 Stage III: 89/312 Stage IV: 64/312 Stage V: 30/312 | 183/327 | NR |
| 2015, Cunha [46] | L-AMB (29) | $\bar{x}$: 68.3 [34–85] (29) | 20:9 | NR | Nose: 24/29 (82.8%). Palate: 10/29 (34.5%). Uvula: 6/29 (20.7%). Gum: 3/29 (10.3%). Jugal mucosa: 2/29 (6.9%). Pharynx: 1/29 (3.4%). Lip: 1/29 (3.4%). | NR | 10 / 29 | NR |
| 2014, Bustos [32] | MA (10) | $\bar{x}$: 54 ± 12 (10) | 7:3 | $\bar{x}$: 272.4 ± 124.8 | NR | According to "mucosal severity score" of Soto, 2007 [48]: $\bar{x}$ = 20 (±3) | NR | * *L. Braziliensis*: 2/19 * *L. amazonensis*: 1/19 |
| | MF (9) | $\bar{x}$: 38 ± 20 (9) | 7:2 | $\bar{x}$: 130.9 ± 168 | | $\bar{x}$ = 17 (±4) | | |
| 2014, Rocio [47] | L-AMB (16) | $\bar{x}$: 59.6 [26–84] (16) | 13:3 | > 5 | NR (The nose was the most common site of mucosal lesions, followed by the pharynx, oral cavity and larynx) | NR | NR | NR |

(*Continued*)

**Table 2.** (Continued)

| Year, Author [reference] | Treatment (N) | Age ($\bar{x}$ or $M_d$ ± SD [Min-Max]) years (n) | Gender (male: female) | Duration of symptoms before therapy (months ± SD) [Min-Max] (n) | Mucosal lesion site: n/N | ML clinical classification: n/N | Previous CL: n/N | Leishmania species: n/N |
|---|---|---|---|---|---|---|---|---|
| 2009, Amato [33] | c-AMB (9) d-AMB (17) L-AMB (4) Intraconazole (15) MA (73) PENT (22) | $M_d$: 57.5 [13–90] (140) | 94:46 | NR | Septal/Nasal: 92/140 Palate: 3/140 Larynx: 9/140 Septal/Nasal with palatal: 23/140 Septal/Nasal with Larynx: 5/140 Septal/Nasal + palate +Larynx: 8/140 | According to Llanos-Cuentas et al, 1997 [42]. Mild: 83/140; moderate: 39/140; severe: 18/140 | 73/140 | NR |
| 2009, Soto [52] | MF (21) | $\bar{x}$: 36 (21) | 15:6 | $\bar{x}$: 14 [4–20] | Only nasal mucosa: 12/15 (cured patients) and 4/5 (failed patients) Nasal mucosa + palate: 1/15 (cured patients) Nasal mucosa + palate, pharynx and/or larynx: 2/15 (cured patients) Only palate: 1/5 (failed patients) | Entrance mucosal severity scores according to "mucosal severity score" of Soto, 2007 [48]: Cured patients (5/21): $\bar{x}$: 8.7 (range = 3–24) Failed patients (6/21): $\bar{x}$: 6 (range = 5–9). | NR | NR |
| 2007, Llanos-Cuentas [50] | AS (21) | $\bar{x}$: 32.6 ± 8.4 (21) | 21:0 | $\bar{x}$: 43.3 ± 52.2 | NR | According to Llanos-Cuentas et al, 1997 [42]: Moderate: 38/38 | NR | NR |
| | MA (17) | $\bar{x}$: 33.2 ± 8.3 (17) | 17:0 | $\bar{x}$: 33.2 ± 26.3 | | | | |
| 2007, Machado [34] | MA (12) | $\bar{x}$: 42 ± 14 (12) | 11:1 | $\bar{x}$: 50 ± 79 | NR | Severe mucosal leishmaniasis (defined as the presence of deep mucosal ulcers and/or septal infiltration or perforation): 23/23 | 9 / 12 | NR |
| | MA + pentoxifylline (11) | $\bar{x}$: 37 ± 15 (11) | 8:3 | $\bar{x}$: 18 ± 36 | | | 7 / 11 | |
| 2007, Soto [48] | MF (78) | $\bar{x}$: 40 ± 16 (78) | 57:21 | $\bar{x}$: 60 ± 60 | Nasal skin + nasal mucosa (mild disease): 40/78 Palate + pharynx + larynx (extensive disease): 38/78 (only Pharynx: 2/38; palate + pharynx: 2/38; pharynx + larynx: 1/38; only larynx:1/38) | Was defined a scale named "**mucosal severity score**", consisting of the sum of the grades for all lesion sites. At any time point the maximum mucosal severity score with which a patient could present was 60: 3 points for each of 4 pathological signs (erythema, edema, infiltration, and erosion) at each of the 5 sites (nasal skin, nasal mucosa, palate, pharynx, and larynx): Score before treatment: $\bar{x}$ = 10 ± 7.6 [1–38] in 72 patients | NR | *L. braziliensis*: 7/78 |
| | d-AMB (19) | NR | NR | NR | NR | NR | NR | NR |
| 2006, Bermudez [41] | SSG (18) | $\bar{x}$: 37.1 ± 13.8 (11)* | 11:0 | NR | NR | NR | NR | NR |
| 2005, Name [27] | Sb$^v$ (148) | [1–81] (402)* | 260:142 * | < 6 (22) > 6 (126) | Nasal septum: 142/164 (87%) (Other affected mucosal membranes were those of the oropharynx, nasopharynx, hard and soft palate and nasal mucosa) | NR | NR | NR |

(Continued)

**Table 2.** (Continued)

| Year, Author [reference] | Treatment (N) | Age ($\bar{x}$ or $M_d$ ± SD [Min-Max]) years (n) | Gender (male: female) | Duration of symptoms before therapy (months ± SD) [Min-Max] (n) | Mucosal lesion site: n/N | ML clinical classification: n/N | Previous CL: n/N | Leishmania species: n/N |
|---|---|---|---|---|---|---|---|---|
| 2004, Calvopina [53] | Itraconazole (13) | $\bar{x}$: 42 ± 13.9 [22–67] (13) | 11:2 | $\bar{x}$: 144 [12–300] | Nasal mucosa: 13/13 (100%) Upper lip: 5/13 (38%) Pharynx: 4/13 (31%) | Patients were classified by the severity of their lesions: severe, septal damage/perforation plus lesions in more than one mucosal site; moderate, lesions in more than one mucosal site without septal damage/perforation; mild, lesions limited to a single mucosal site.—Severe: 7/13 (54%)—Moderate: 4/13 (31%)—Mild: 2/13 (15%) | 12/13 | *Leishmania (Viannia) braziliensis*: 2 patients *Leishmania (Viannia)*: 7 patients NR: 4 patients |
| 2000, Oliveira-Neto [39] | MA-LD (31) | $\bar{x}$: 57.6 ± 15.2 (36)* | 22:14 * | $\bar{x}$: 127.6 ± 118.6 | Nasal: 36/36 (100%) Palate: 14/36 (39%) Pharynx: 10/36 (28%) Larynx: 7/36 (19%) Cavum (nasofaringe): 4/36 (11%) Maxillary sinus: 1/36 (3%) | According to Llanos-Cuentas et al, 1997 [42]. The group of patients studied in this paper presented a mild to moderate mucosal disease. | NR | NR |
| 1997, Llanos-Cuentas [42] | SSG, phase 1 (11) | $\bar{x}$: 34.3 ± 6 (11) | 81:0 | $\bar{x}$: 33.6 ± 18.9 | Nose + nasopharynx + Palate: 2/11 Nose + nasopharynx + Palate + epiglottis: 0/11 Nose + nasopharynx + Palate + epiglottis + vocal cords: 9/11 | Severity of symptoms (mild, those symptoms confined to the nose; **moderate**, odynophagia, dysphonia, and/or mild respiratory distress; and **severe**, odynophagia, dysphonia, and severe respiratory distress): mild = 0/11; moderate = 0/11; severe = 11/11. | 81/81 | *Leishmania braziliensis* complex: 55/81 |
| | SSG, phase 2 (30) | $\bar{x}$: 33 ± 7.9 (30) | | $\bar{x}$: 34.2± 28.1 | Nose + nasopharynx + Palate: 8/30 Nose + nasopharynx + Palate + epiglottis: 16/30 Nose + nasopharynx + Palate + epiglottis + vocal cords: 6/30 | Mild = 8/30; Moderate = 22/30; Severe = 0/30. | | |
| | SSG + allopurinol, phase 1 (11) | $\bar{x}$: 36.1 ± 8.6 (11) | | $\bar{x}$: 34.9 ± 25.6 | Nose + nasopharynx + Palate: 0/11 Nose + nasopharynx + Palate + epiglottis: 0/11 Nose + nasopharynx + Palate + epiglottis + vocal cords: 11/11 | Mild = 0/11; Moderate = 0/11; Severe = 11/11. | | |
| | SSG + allopurinol, phase 2 (29) | $\bar{x}$: 32.8 ± 8.9 (29) | | $\bar{x}$: 37.9 ± 41.2 | Nose + nasopharynx + Palate: 8/29 Nose + nasopharynx + Palate + epiglottis: 18/29 Nose + nasopharynx + Palate + epiglottis + vocal cords: 3/29 | Mild = 14/29; Moderate = 15/29; Severe = 0/29. | | |

(*Continued*)

**Table 2.** (Continued)

| Year, Author [reference] | Treatment (N) | Age ($\bar{x}$ or $M_d$ ± SD [Min-Max]) years (n) | Gender (male: female) | Duration of symptoms before therapy (months ± SD) [Min-Max] (n) | Mucosal lesion site: n/N | ML clinical classification: n/N | Previous CL: n/N | Leishmania species: n/N |
|---|---|---|---|---|---|---|---|---|
| 1996, Romero [51] | AS (13) | $\bar{x}$: 36.9 (13) | 11:2 | $\bar{x}$: 18.6 (13) | Nasal lesion*: 20/21 (95%), with 18/21 presenting septum ulcer or perforation. AS = septum perforation: 3/13 (23%) | Single lesion: 11/13 Multiple lesions: 2/13 | 19 / 21* | NR |
| 1995, Oliveira [36] | MA (51) | NR | 60:17 * | NR | Nasal mucosa: 74/77 (96.1%) | NR | 67/77 | NR |
| 1994, Franke [43] | SSG, 28 days (20) | $\bar{x}$: 33.7 ± 7.3 [24–47] (20) | 40:0 | $\bar{x}$: 34.8 ± 25.2 [3.6–102] (20) | NR | NR | 19/20 | L. braziliensis: 35/40 |
| | SSG, 40 days (20) | $\bar{x}$: 30.7 ± 6.3 [22–42] (20) | | $\bar{x}$: 34.8 ± 31.2 [2.4–120] (20) | | | 18/20 | |
| 1993, Zocoli [26] | ketoconazole (25) Sbᵛ (98) Sbᵛ+ sulfa (180) | [0–10] (8) [10–20] (35) [20–30] (72) [30–40] (97) [40–50] (84) | NR | NR | Nose: 257/303 (85%) | NR | 46/303 | NR |
| 1991, Kopke [37] | MA (17) | NR | NR | NR | NR | NR | NR | NR |
| 1991, Saenz [44] | SSG (16) | [18 – 33] (13) [53 – 74] (3) | 7:9 | $\bar{x}$: 14 [10–36] (14) 168 (1) 360 (1) | Septum: 10/16 (63%) Septum turbinate: 4/16 (25%) Nasal skin: 2/16 (13%) Turbinate: 1/16 (6%) Alae: 1/16 (6%) Cheek: 1/16 (6%) Thigh: 1/16 (6%) | Mild disease (no perforation): 15/16 Moderate disease (with septal perforation): 1/16 | 7/16 | Leishmania panamensis: 3/16 |
| 1990, Franke [45] | SSG (29) | $\bar{x}$: 32 ± 9 [20–58] (29) | 27:2 | $\bar{x}$: 36 ± 33.6 [2.4–168] (29) | Nasal mucosa: 29/29 Palate: 14/29 Pharynx: 12/29 Epiglottis: 11/29 Nasal skin: 4/29 Lip: 3/29 Uvula: 2/29 Larynx: 2/29 Vocal cords: 1/29 | Mild (confined to the nasal mucosa): 6 /29 (21%) Moderate (limited to the nose but with septal perforation): 2/29 (7%) Severe (involving the oral cavity as well as the nose): 21/29 (72%) | 28/ 29 | L. braziliensis braziliensis: 22/ 29 |
| 1989, Sampaio [29] | MA (26) Sbᵛ (12) SSG (18) | NR | NR | NR | NR | NR | NR | NR |
| 1960, Sampaio [49] | d-AMB (11) | $\bar{x}$: 34.6 ± 14.8 [10–55] (11)<sup>&</sup> | 10<sup>&</sup>:1 | $\bar{x}$: 98.5 ± 68.3 [4–216] (11) | Nasal mucosa: 4/11 Pharynx: 1/11 Larynx: 7/11 Nasobucopharynx: 3/11 Nasopharynx: 2/11 Lip: 2/11 | NR | NR | NR |

**&**: A case with a lip lesion (table) not reported in the text was considered as ML.

*: the number of participants reporting this information is different from the number of participants reporting a cure rate. **ABLC**: Amphotericin B lipid complex. **AS**: Aminosidine sulphate. **c-AMB**: Amphotericin B colloidal dispersion. **d-AMB**: Deoxycholate amphotericin B. **L-AMB**: Liposomal amphotericin B. **MA**: Meglumine antimonate. **MA-LD**: Meglumine antimonate low dose. **Max**: maximum. **M$_d$**: median. **MF**: Miltefosine. **Min**: minimum. **ML**: mucosal leishmaniasis; **n**: number of cases. **N**: total of patients. **NR**: no reported. **OS**: Other states. **PENT**: pentamidine. **RJ**: Rio de Janeiro. **Sbᵛ**: Antimonial pentavalent. **SD**: standard deviation. **SSG**: Sodium stibogluconate. $\bar{x}$: mean.

## Outcomes

The cure and relapse rates described by each study at different times after the start of treatment are shown in **Table 3**. Taking as a model the outcomes harmonization for CL studies [14,55], it was possible to gather 12 studies [30,31,34,38,40,41,44,45,49,51,53] reporting cure at D90 (meaning cure assessment between 60 and 135 days from the beginning of treatment), 10 studies [26,29–31,34,37,44–46,53] presenting cure at D180 (cure assessment between 136 and 270 days from the beginning of treatment) and 16 studies [27–29,31–33,36,37,41–45,47,48,51,52] presenting cure at D360 (cure assessment between 271 and 390 from the beginning of treatment).

The pooled cure rate based on the latest cure assessment reported in the original studies was calculated grouping all study arms addressing the same intervention as shown in **Figs 2–8**. The overall cure rate including patients treated with all derivatives and doses of pentavalent antimonials was 63.3% (CI: 56.2–69.8%; $I^2$ = 72.4) (Fig 2). No significant difference was observed between patients treated with standard MA therapeutic doses (15–20 mg/kg for 10–30 days) in comparison to MA low doses (5 mg/kg/day): 65.1% (CI: 52.8–75.6%; $I^2$ = 55.8) and 67.2% (CI: 42.1–85.2%; $I^2$ = 0), respectively. The cure rate of the 299 patients treated with MA derivatives (66.2%, CI: 57.7–73.9%) was significantly higher than that observed for the 156 patients treated with SSG (51.8%, CI: 39.5–64%), p = 0.00. The combination of MA with pentoxifylline apparently increased the cure rate based on an indirect comparison between patients treated with and without this adjuvant drug (77.4% and 65.1%, respectively, p = 0.00). On the other hand, the cure rate of patients treated with other combinations (MA plus allopurinol or sulfa) was similar to that observed for MA standard dose therapy (64.0% and 65.1%, respectively).

The cure rates for patients treated with the different amphotericin formulations were pooled into two groups: the lipidic amphotericin B formulations (79.4%, CI: 69.7–86.5%; $I^2$ = 48.8) and deoxycholate amphotericin B formulation (39.5%, CI:16.4–68.5%; $I^2$ = 71), (p = 0.0001) (**Figs 3 and 4**). The cure rates for the other interventions were: 83.3% (CI: 57.8–94.8%; $I^2$ = 42) for pentamidine (**Fig 5**); 65.2% (CI: 56.4–73%; $I^2$ = 0) for miltefosine (**Fig 6**); 53.3% (CI:28.9–76.2%; $I^2$ = 80) for imidazoles (**Fig 7**) and 11.9% (CI: 0.8–69.8%; $I^2$ = 72) for aminosidine (**Fig 8**). Except for the studies addressing miltefosine, for all other intervention groups, high intragroup heterogeneity was observed.

Two RCTs [30,32] compared the same interventions, MA and miltefosine, allowing a direct comparison in the meta-analysis (Fig 9). Involving 57 patients and considering the cure rate assessed in setpoints as close as possible, no difference was observed between these interventions (OR: 1.2; 0.43–3.49, $I^2$ = 0).

## Adverse events

The occurrence of AEs was reported in 21 of the 27 studies (**S3 Table**). For most primary studies, the reporting of AE did not follow a systematic and active methodology. The analysis was generally incomplete (causality and expectation were not assessed), and different AE designations and toxicity classification systems were used. Only four studies explicitly reported the use of an AE classification system: three of them adopted the Common Terminology Criteria for Adverse Events (CTCAE) [32,48,52], and one study used the Division of AIDS Table for Grading of Severity of Adult and Paediatric Adverse Events [38]. Even without adopting an EA classification system, the suspension of treatment due to the occurrence of an AE was reported in at least 10 articles [29,30,34,37,41–45,47].

Considering only studies that reported AEs, we identified 1,401 AEs among 1,008 treated patients. For most studies, AEs were reported in absolute numbers, and not as the proportion

**Table 3. Outcomes: cure and relapse rates according to the intent to treatment approach.**

| Year, Author [reference] | Treatment (N) | D90 Cure rate (%) | D180 Cure rate (%) | D360 Cure rate (%) | Latest cure rate (%) | Relapse rate (%) |
|---|---|---|---|---|---|---|
| 2019, Sampaio [30] | MA (18) | 7/18 (39%) | 9/18 (50%) | NR | 12/18 (67%), 4 years after the end of treatment | NR |
| | MF (20) | 11/20 (55%) | 11/20 (55%) | NR | 16/20 (80%), 4 years after the end of treatment | NR |
| 2019, Santos& [28] | ABLC (13) | NR | NR | 6/13 (46.2%) | 6/13 (46.2%) | 12/71 (16.9%) |
| | d-AMB (14) | | | 2/14 (14.3%) | 2/14 (14.3%) | |
| | L-AMB (32) | | | 26/32 (81.3%) | 26/32 (81.3%) | |
| | Itraconazole (10) | | | 4/10 (40%) | 4/10 (40%) | |
| | PENT (11) | | | 8/11 (72.7%) | 8/11 (72.7%) | |
| | Sb$^v$ (25) | | | 14/25 (56%) | 14/25 (56%) | |
| 2018, Cataldo [38] | MA-LD (27) | 18/27 (66.7%) | NR | NR | 18/27 (66.7%) | NR |
| 2018, Pedras [31] | Fluconazole (9) | 2/9 (22.2%) | 3/9 (35%) | 2/9 (22%) | 2/9 (22.2%), at the one-year follow-up visit. | 0 |
| | L-AMB (9) | 6/9 (67%) | 7/9 (78%) | 7/9 (77.8%) | 7/9 (77.8%), at the one-year follow-up visit. | 1/9 (11%) |
| | MA (17) | 6/17 (35%) | 12/17 (78%) | 14/17 (82.4%) | 14/17 (82.4%), at the one-year follow-up visit. | 0 |
| 2017, Cincurá [40] | MA or MA + pentoxifylline (251) | 179/251 (71.3%) | NR | NR | 179/251 (71.3%) | 25/251 (10%) |
| 2015, Cunha [46] | L-AMB (29) | NR | 27/29 (93,1%) | NR | 27/29 (93,1%) | 2/29 (6.9%) |
| 2014, Bustos [32] | MA (10) | NR | NR | 5*/10 (50%) | 5*/10 (50%), one year after conclusion of therapy. | NR |
| | MF (9) | NR | NR | 5*/9 (55.6%) | 5*/9 (55.6%), one year after conclusion of therapy. | NR |
| 2014, Rocio [47] | L-AMB (16) | NR | NR | 14/16 (87.5%) | 14/16 (87.5%), one year after conclusion of therapy. | NR |
| 2009, Amato [33] | c-AMB (9) | NR | NR | 8/9 (88.9%) | 8/9 (88.9%), one year after conclusion of therapy. | 0/9 (0%) |
| | d-AMB (17) | NR | NR | 5/17 (29.4%) | 5/17 (29.4%), one year after conclusion of therapy. | 0/17 (0%) |
| | L-AMB (4) | NR | NR | 4/4 (100%) | 4/4 (100%), one year after conclusion of therapy. | 1/4 (25%) |
| | Itraconazole (15) | NR | NR | 11/15 (73.3%) | 11/15 (73.3%), one year after conclusion of therapy. | 2/15 (13.3%) |
| | MA (73) | NR | NR | 58/73 (79.5%) | 58/73 (79.5%), one year after conclusion of therapy. | 13/73 (17.8%) |
| | PENT (22) | NR | NR | 20/22 (90.9%) | 20/22 (90.9%), one year after conclusion of therapy. | 5/22 (22.7%) |
| 2009, Soto [52] | MF (21) | NR | NR | 14*/21 (66.7%) | 14*/21 (66.7%), by 12 months of follow-up. | NR |
| 2007, Llanos-Cuentas [50] | AS (21) | NR | NR | 0/21 (0%) | 0/21 (0%), 1 year after finishing treatment. | NR |
| | MA (17) | NR | NR | 8/17 (47.1%) | 8/17 (47.1%), 1 year after finishing treatment. | NR |
| 2007, Machado [34] | MA (12) | 5/12 (42%) | 7/12 (58.3%) | NR | 7/12 (58.3%) | 0/12 (0%), at least 2 years after treatment cessation |
| | MA + pentoxifylline (11) | 9/11 (82%) | 11/11 (100%) | NR | 11/11 (100%) | 0/11 (0%), at least 2 years after treatment cessation |

(*Continued*)

**Table 3.** (Continued)

| Year, Author [reference] | Treatment (N) | D90 Cure rate (%) | D180 Cure rate (%) | D360 Cure rate (%) | Latest cure rate (%) | Relapse rate (%) |
|---|---|---|---|---|---|---|
| 2007, Soto [48] | d-AMB (19) | NR | NR | 7/19 (36.8%) | 7/19 (36.8%), 12 months after the end of therapy. | NR |
| | MF (78) | NR | NR | 49*/78 (62.8%) | 49*/78 (62.8%), 12 months after the end of therapy. | NR |
| 2006, Bermudez [41] | SSG (18) | 8/18 (44.4%) | NR | 8/18 (44.4%) | 8/18 (44.4%) | NR |
| 2005, Name [27] | Sb$^v$ (148) | NR | NR | 103/148 (69.6%) | 103/148 (69.6%) | NR |
| 2004, Calvopina [53] | Itraconazole (13) | 3/13 (23.1%) | 3/13 (23.1%) | NR | 3/13 (23.1%), 6 months after treatment. | NR |
| 2000, Oliveira-Neto [39] | MA-LD (31) | NR | NR | NR | 21/31 (67.7%), after 45 days of therapy. | 0/31 (0%), after one to seven years of follow-up. |
| 1997, Llanos-Cuentas [42] | SSG (41) | NR | NR | 23/41 (56.1%) | 23/41 (56.1%), 12 month follow-up period. | 15/41 (36.6%) |
| | SSG + allopurinol (40) | NR | NR | 14/40 (35%) | 14/40 (35%), 12 month follow-up period. | 18/40 (45%) |
| 1996, Romero [51] | AS (13) | 7/13 (53.8%) | NR | 4/13 (30.8%) | 4/13 (30.8%), at 1 year of follow-up. | NR (reports 4 relapses, but does not specify the group) |
| 1995, Oliveira [36] | MA (51) | NR | NR | 42/51 (82.4%) | 42/51 (82.4%), at least 1 year of follow-up. | 6/51 (11,8%) |
| 1994, Franke [43] | SSG, 28 days (20) | 2/20 (10%) | NR | 10/20 (50%) | 10/20 (50%) | NR (reports 12 lesions relapsed, but does not specify the number of patients or group) |
| | SSG, 40 days (20) | 2/20 (10%) | NR | 12/20 (60%) | 12/20 (60%) | |
| 1993, Zocoli [26] | ketoconazole (25) | NR | 23/25 (92%) | NR | 23/25 (92) | 2/25 (8%) |
| | Sb$^v$ (98) | NR | 72/98 (73.5%) | NR | 72/98 (73.5%) | 26/98 (26.5%) |
| | Sb$^v$ + sulfa (180) | NR | 150/180 (83.3%) | NR | 150/180 (83.3%) | 30/180 (16.7%) |
| 1991, Kopke [37] | MA: 14 mg/kg/day (10) | NR | 0/10 (0%) | 1/10 (10%) | 4/10 (40%), after 2 years of follow-up | NR |
| | MA: 28 mg/kg/day (7) | NR | 0/7 (0%) | 2 /7 (28.6%) | 4/7 (57%), after 2 years of follow-up | NR |
| 1991, Saenz [44] | SSG (16) | 8/16 (50%) | 11/16 (69%) | 9/16 (56.3%) | 9/16 (56.3%), 12 months after the end therapy | 4/16 (25%) |
| 1990, Franke [45] | SSG (29) | 12/29 (41.4%) | 10/19 (34.5%) | 8 /29 (27.6%) | 8 / 29 (27.6%) | 6/29 (20.7%) |
| 1989, Sampaio [29] | MA (26) | NR | 8/26 (30.8%) | NR | 15/26 (57.7%), at an average time of two and a half years of follow-up | NR |
| | Sb$^v$ (12) | NR | 11/12 (91.7%) | NR | 11/12 (91.7%), at an average time of 8 months of follow-up | NR |
| | SSG (18) | NR | NR | 11/18 (61.1%) | 11/18 (61.1%), at an average time of one year of follow-up | NR |
| 1960, Sampaio [49] | d-AMB (11) | 10/11 (90.9%) | NR | NR | 10/11 (90.9%) | 1/11 (9.1%), case 4 had relapsed after seven months of therapy. |

**&**: for all cure rates, the information available was based on the number of cures in relation to the number of treatments performed, which does not rule out that the same patient may have received more than one treatment.

*Unlike the criterion adopted by the author in the study, only patients with 100% epithelialization were counted to maintain alignment with the definition of cure adopted in the other studies. **ABLC**: Amphotericin B lipid complex. **AS**: Aminosidine sulphate. **c-AMB**: Amphotericin B colloidal dispersion. **d-AMB**: Deoxycholate amphotericin B. **L-AMB**: Liposomal amphotericin B. **MA**: Meglumine antimonate. **MA-LD**: Meglumine antimonate low dose. **MF**: Miltefosine. **N**: number of patients. **NR**: not reported. **PENT**: pentamidine. **Sb$^v$**: Antimonial pentavalent. **SSG**: Sodium stibogluconate.

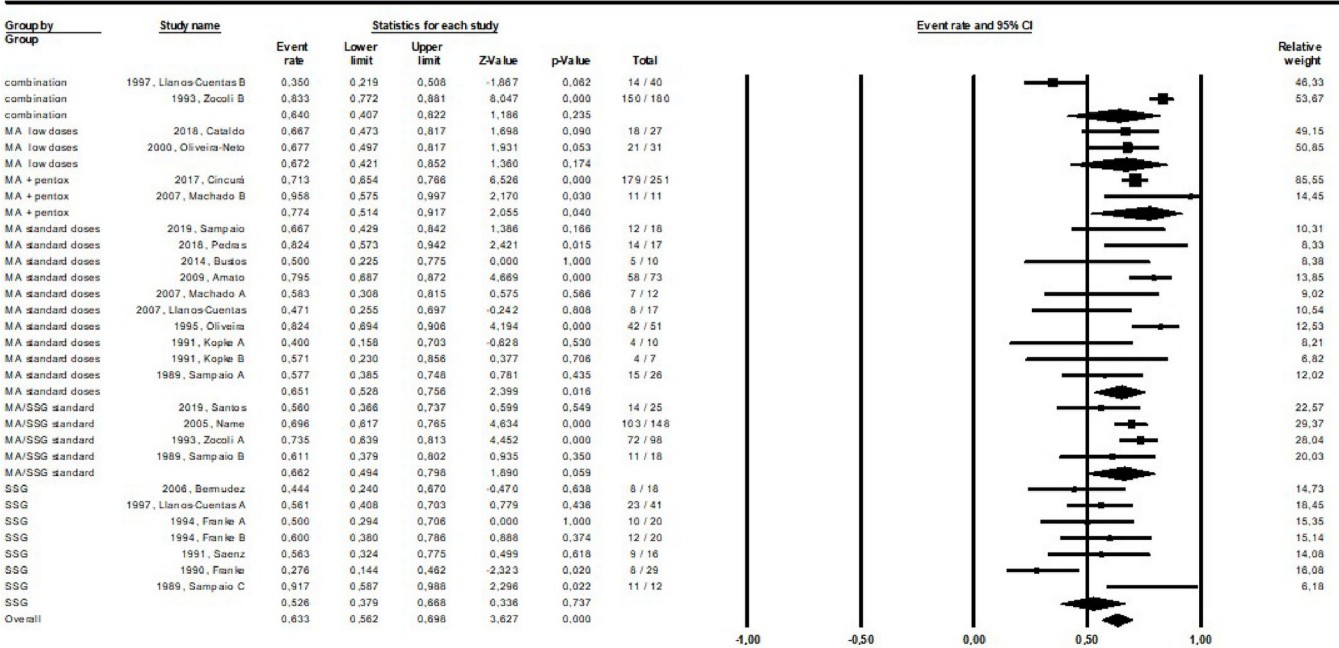

**Fig 2. Pooled cure rate of pentavalent antimonials using a mixed effects analysis. MA:** Meglumine antimonate. **Pentox**: pentoxifylline. **SSG**: Sodium stibogluconate. Different interventions in the same study were indicated by letters.

| Group by Group | Study name | Event rate | Lower limit | Upper limit | Z-Value | p-Value | Total | Event rate and 95% CI |
|---|---|---|---|---|---|---|---|---|
| Colloidal disper | 2009, Amato A | 0,889 | 0,500 | 0,985 | 1,961 | 0,050 | 8 / 9 | |
| Colloidal disper | | 0,889 | 0,500 | 0,985 | 1,961 | 0,050 | | |
| Lipid complex | 2019, Santos B | 0,462 | 0,224 | 0,718 | -0,277 | 0,782 | 6 / 13 | |
| Lipid complex | | 0,462 | 0,224 | 0,718 | -0,277 | 0,782 | | |
| Liposomal | 2019, Santos A | 0,813 | 0,641 | 0,913 | 3,238 | 0,001 | 26 / 32 | |
| Liposomal | 2018, Pedras | 0,778 | 0,421 | 0,944 | 1,562 | 0,118 | 7 / 9 | |
| Liposomal | 2015, Cunha | 0,931 | 0,762 | 0,983 | 3,552 | 0,000 | 27 / 29 | |
| Liposomal | 2014, Rocio | 0,875 | 0,614 | 0,969 | 2,574 | 0,010 | 14 / 16 | |
| Liposomal | 2009, Amato B | 0,900 | 0,326 | 0,994 | 1,474 | 0,140 | 4 / 4 | |
| Liposomal | | 0,852 | 0,758 | 0,913 | 5,661 | 0,000 | | |
| Overall | | 0,794 | 0,697 | 0,865 | 5,150 | 0,000 | | |

**Fig 3. Pooled cure rate including all patients treated with lipid formulations of amphotericin B in the reviewed studies using a mixed effect analysis. Colloidal disper:** colloidal dispersion. **Liposomal**: Liposomal amphotericin B. Different interventions in the same study were indicated by letters.

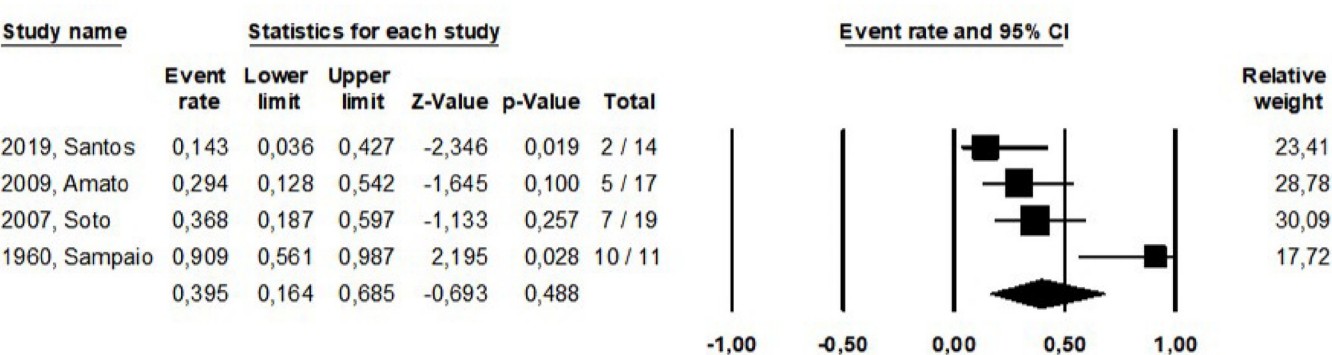

**Fig 4. Pooled cure rate including all patients treated with deoxycholate amphotericin in the reviewed studies.**

of treated patients with a given intervention. Few authors described serious or severe AEs; however, it was possible to identify treatment discontinuation due to AE for 37 patients (3.6%): 6 patients in therapy with MA (1 cardiac event, 2 laboratory alterations, 2 leukopenia, and 1 Jarisch-Herxheimer reaction); 1 patient under MF therapy (abdominal pain and elevated serum amylase); 5 patients using L-AMB (acute renal failure); 3 patients using d-AMB (vomiting or increased creatinine level); 10 patients using SSG (5 thrombocytopenia, 3 liver function test abnormality, 1 thrombocytopenia and 1 aspartate aminotransferase increase); 11 patients using SSG plus allopurinol (8 thrombocytopenia, 2 bradycardia, 1 liver function test abnormality, and 1 infectious condition), and 1 patient using AS (hearing loss for high-frequency sounds) (**S3 Table**).

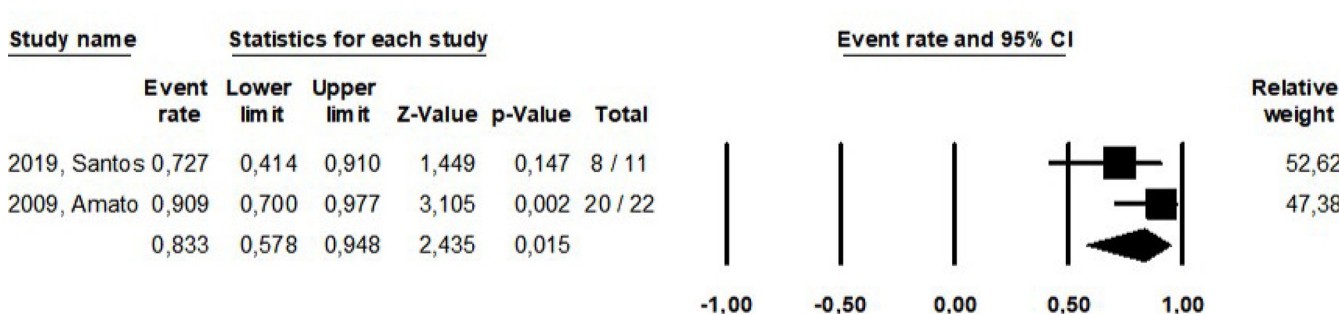

**Fig 5. Pooled cure rate including all patients treated for ML with pentamidine in the reviewed studies.**

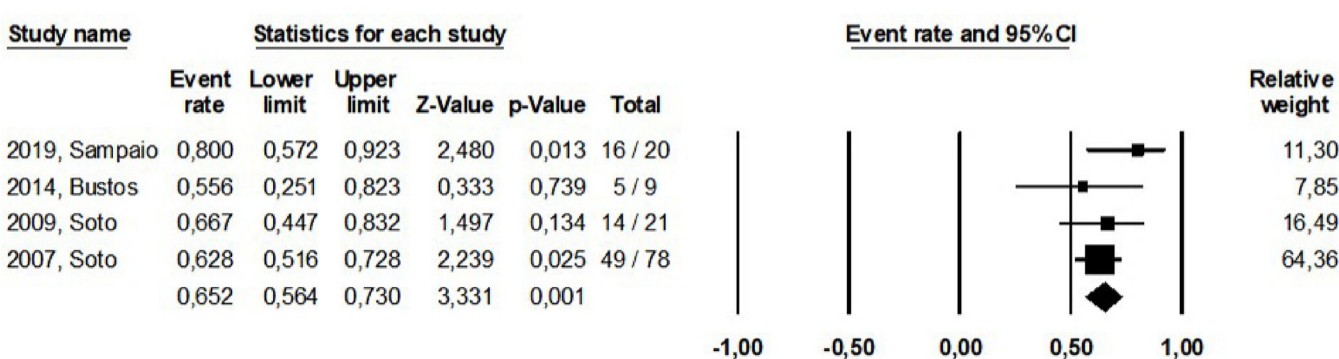

**Fig 6. Pooled cure rate including all patients treated for ML with miltefosine in the reviewed studies.**

The detailed description of AEs by SOC for each intervention is presented in **Table 4**. Overall, 64% of the AEs occurred among patients treated with antimony derivatives, and the therapy received by 45% (744 of 1,666) of patients gathered in this review. Among patients treated with antimonial drugs, musculoskeletal and connective tissue disorders were the AEs most identified, followed by laboratory abnormalities, named the "investigations" SOC term (**S4 Table**).

Among the 173 patients treated with one of the amphotericin formulations, we identified 182 AEs, which corresponds to 13% of the events accounted for in this review. "Injury, poisoning and procedural complications", such as infusion-related reactions and "Renal and urinary disorders" were the AEs most reported for both groups of amphotericin B formulations (lipid formulations and deoxycholate).

The most frequent AEs reported among patients treated with AS were "General disorders and administration site conditions" (such as fever, chest pain, chills, malaise, etc.), while gastrointestinal alterations were the most described among those treated with MF.

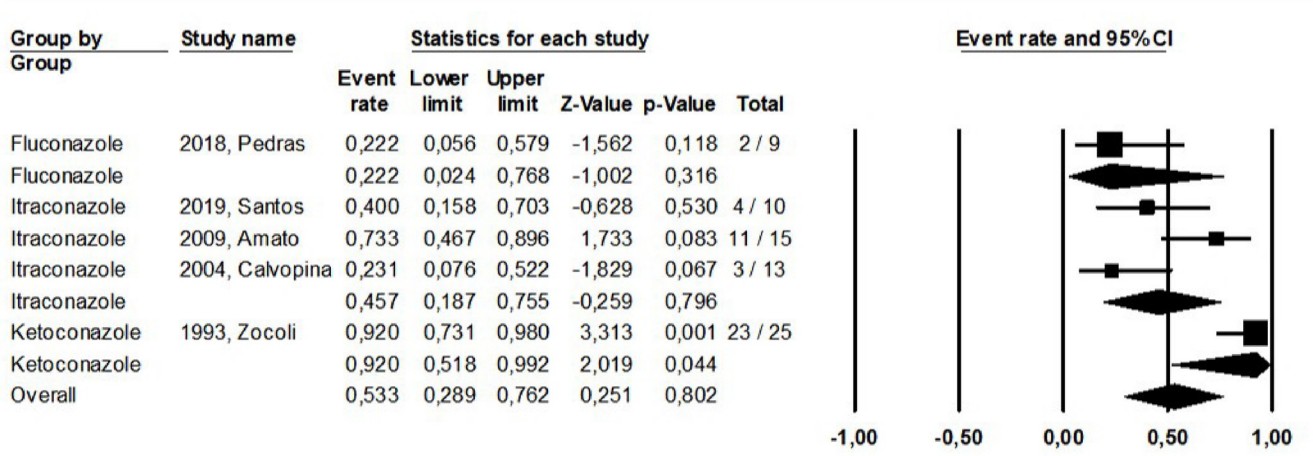

**Fig 7. Pooled cure rate including all patients treated for ML with imidazole drugs in the reviewed studies using a mixed effect analysis.**

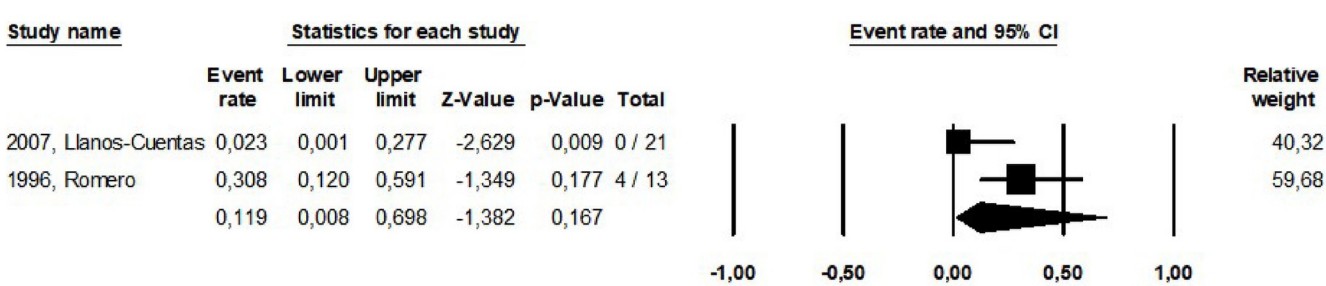

**Fig 8. Pooled cure rate including all patients treated for ML with aminosidine in the reviewed studies.**

## Risk of bias in included studies

The risk of bias assessment is presented in **Figs 10** and **S1.** For RCTs, overall, the risk of bias was considered moderate. The highest risk domain was outcome assessment (mainly due to lack of blinding), followed by the absence of allocation concealment during randomization. As expected, for cohort and nonrandomized studies, the major concern, present for all studies, was the absence of comparator or comparability between groups with different interventions, followed by risk of bias in the selection of the population. Finally, for retrospective studies, the risk of bias was present in all evaluated domains, with greater intensity for the ascertainment bias.

## Discussion

The main observation of this review is the identification of a relatively small number of published therapeutic interventions for ML and, in general, a small number of clinical trials addressing the disease. As expected, antimony derivatives account for most of the treatments gathered here, including the two pentavalent antimony derivatives, meglumine and

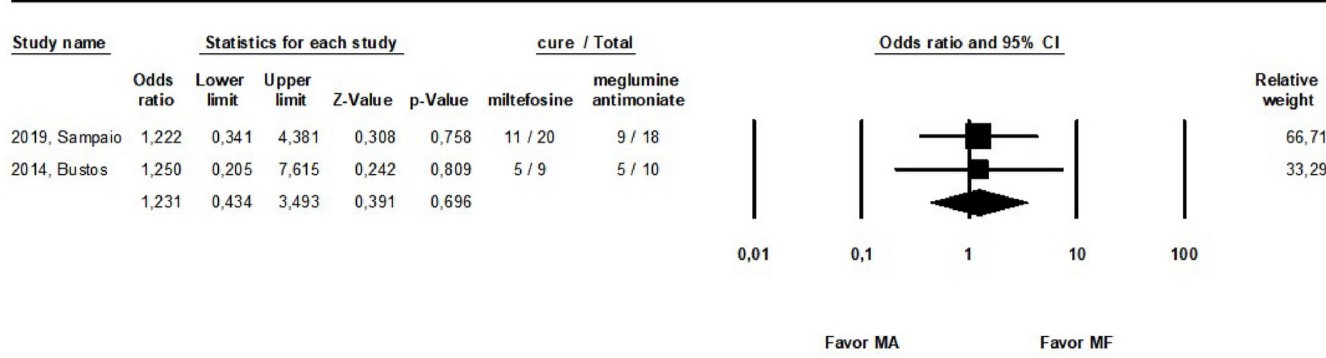

**Fig 9. Meta-analyzis of studies directly comparing meglumine antimoniate and miltefosine for the treatment of ML.**

**Table 4. System Organ Classes for adverse events according to therapy.**

| System Organ Classes term | Number of AE reported | % AE by therapy |
|---|---|---|
| **Sbv** | **904** | **100.0%** |
| Musculoskeletal and connective tissue disorders | 283 | 31.3% |
| **Investigations (laboratorial abnormalities)** | **218** | **24.1%** |
| General disorders and administration site conditions | 87 | 9.6% |
| Gastrointestinal disorders | 84 | 9.3% |
| Nervous system disorders | 69 | 7.6% |
| Metabolism and nutrition disorders | 42 | 4.6% |
| Blood and lymphatic system disorders | 37 | 4.1% |
| NR | 19 | 2.1% |
| Skin and subcutaneous tissue disorders | 15 | 1.7% |
| Respiratory, thoracic and mediastinal disorders | 11 | 1.2% |
| Injury, poisoning and procedural complications | 11 | 1.2% |
| Vascular disorders | 8 | 0.9% |
| Infections and infestations | 7 | 0.8% |
| Cardiac disorders | 4 | 0.4% |
| Immune system disorders | 4 | 0.4% |
| Psychiatric disorders | 4 | 0.4% |
| Ear and labyrinth disorders | 1 | 0.1% |
| **SSG + allopurinol** | **191** | **100.0%** |
| Musculoskeletal and connective tissue disorders | 53 | 27.7% |
| Nervous system disorders | 38 | 19.9% |
| General disorders and administration site conditions | 36 | 18.8% |
| Gastrointestinal disorders | 22 | 11.5% |
| **Investigations** | **13** | **6.8%** |
| Blood and lymphatic system disorders | 12 | 6.3% |
| Metabolism and nutrition disorders | 12 | 6.3% |
| Respiratory, thoracic and mediastinal disorders | 3 | 1.6% |
| Infections and infestations | 2 | 1.0% |
| **L-AMB** | **99** | **100.0%** |
| Injury, poisoning and procedural complications | 31 | 31.3% |
| Renal and urinary disorders | 16 | 16.2% |
| Metabolism and nutrition disorders | 12 | 12.1% |
| General disorders and administration site conditions | 10 | 10.1% |
| Musculoskeletal and connective tissue disorders | 7 | 7.1% |
| Gastrointestinal disorders | 6 | 6.1% |
| Vascular disorders | 6 | 6.1% |
| Nervous system disorders | 5 | 5.1% |
| Cardiac disorders | 3 | 3.0% |
| **Investigations (laboratorial abnormalities)** | **3** | **3.0%** |
| **AS** | **51** | **100.0%** |
| General disorders and administration site conditions | 26 | 51.0% |
| Musculoskeletal and connective tissue disorders | 17 | 33.3% |
| Renal and urinary disorders | 6 | 11.8% |
| Ear and labyrinth disorders | 1 | 2.0% |
| Metabolism and nutrition disorders | 1 | 2.0% |
| **ABLC** | **47** | **100.0%** |
| Injury, poisoning and procedural complications | 10 | 21.3% |

(*Continued*)

**Table 4.** (Continued)

| System Organ Classes term | Number of AE reported | % AE by therapy |
|---|---|---|
| General disorders and administration site conditions | 10 | 21.3% |
| Vascular disorders | 6 | 12.8% |
| Nervous system disorders | 6 | 12.8% |
| Metabolism and nutrition disorders | 6 | 12.8% |
| Gastrointestinal disorders | 5 | 10.6% |
| Musculoskeletal and connective tissue disorders | 2 | 4.3% |
| Cardiac disorders | 1 | 2.1% |
| Renal and urinary disorders | 1 | 2.1% |
| **MF** | **43** | **100.0%** |
| Gastrointestinal disorders | 40 | 93.0% |
| **Investigations (laboratorial abnormalities)** | **2** | **4.7%** |
| Renal and urinary disorders | 1 | 2.3% |
| **d-AMB** | **36** | **100.0%** |
| Renal and urinary disorders | 8 | 22.2% |
| Injury, poisoning and procedural complications | 7 | 19.4% |
| Metabolism and nutrition disorders | 7 | 19.4% |
| General disorders and administration site conditions | 4 | 11.1% |
| Gastrointestinal disorders | 4 | 11.1% |
| Vascular disorders | 3 | 8.3% |
| Nervous system disorders | 3 | 8.3% |
| Pregnancy, puerperium and perinatal conditions | 0 | 0.0% |
| **Investigations (laboratorial abnormalities)** | **0** | **0.0%** |
| **Pentamidine** | **20** | **100.0%** |
| Injury, poisoning and procedural complications | 5 | 25.0% |
| Metabolism and nutrition disorders | 5 | 25.0% |
| Vascular disorders | 4 | 20.0% |
| Gastrointestinal disorders | 3 | 15.0% |
| Nervous system disorders | 2 | 10.0% |
| General disorders and administration site conditions | 1 | 5.0% |
| **MA + pentoxifylline** | **7** | **100.0%** |
| Gastrointestinal disorders | 5 | 71.4% |
| Nervous system disorders | 1 | 14.3% |
| Musculoskeletal and connective tissue disorders | 1 | 14.3% |
| **Fluconazole** | **3** | **100.0%** |
| Metabolism and nutrition disorders | 1 | 33.3% |
| Gastrointestinal disorders | 1 | 33.3% |
| **Investigations (laboratorial abnormalities)** | **1** | **33.3%** |
| **Total** | **1401** | |

ABLC: Amphotericin B lipid complex. **AE:** Adverse Event. **AS:** Aminosidine sulphate. **d-AMB:** Deoxycholate amphotericin B. **L-AMB:** Liposomal amphotericin B. **MA:** Meglumine antimonate. **MF:** Miltefosine. **NR:** No reported. **Sb<sup>v</sup>:** Antimonial pentavalent. **SSG:** Sodium stibogluconate.

stibogluconate, in addition to antimony combinations with sulfa, allopurinol or pentoxifylline, used in several therapeutic regimens (different doses, durations and scheduled administrations). The other therapies retrieved for ML were different formulations of amphotericin B, aminosidine, pentamidine, miltefosine, and imidazoles.

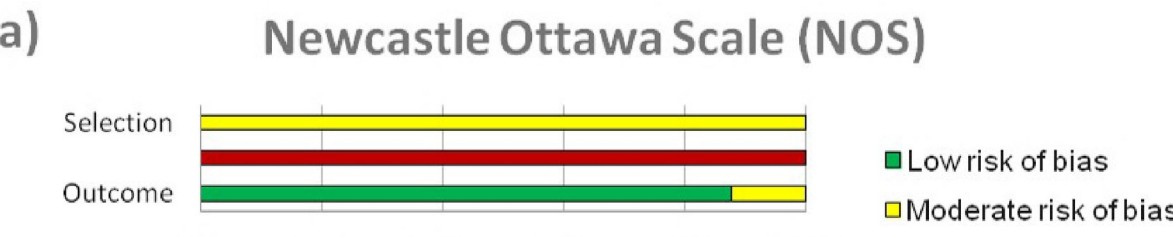

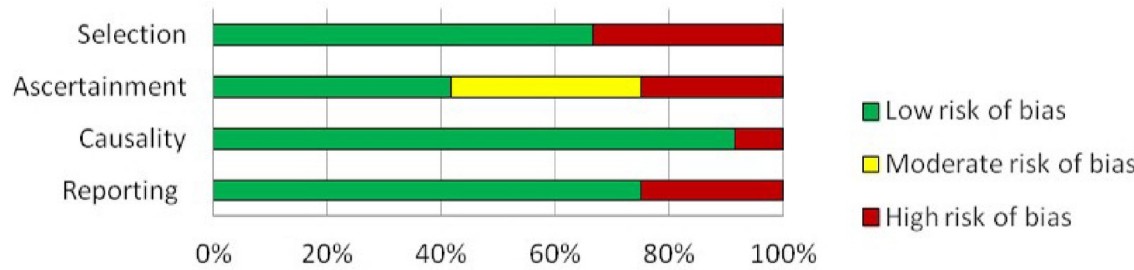

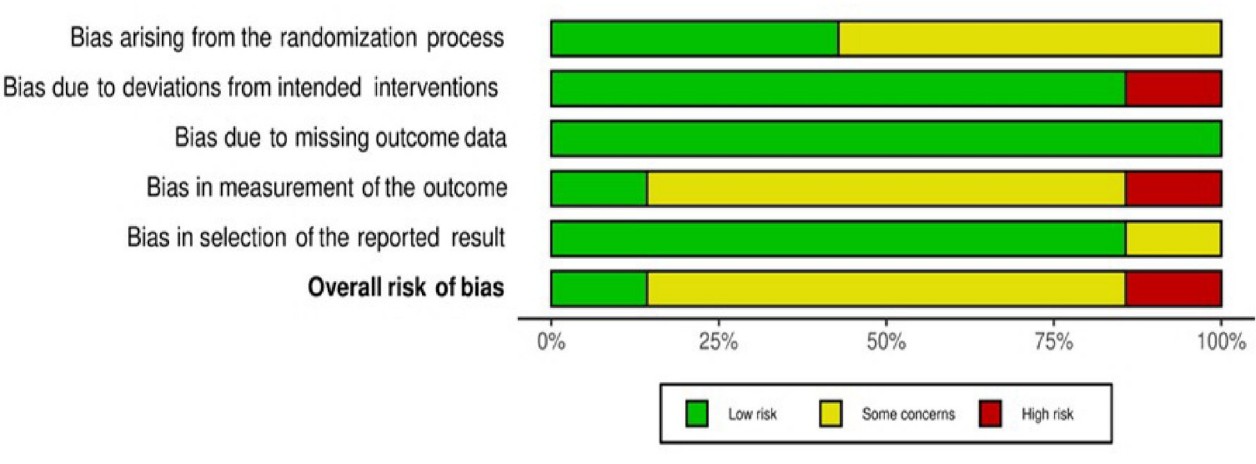

**Fig 10. Risk of bias assessment.**

The absence of controlled trials prevented the performance of a classical meta-analysis with direct comparison between treatments. Furthermore, due to the high heterogeneity and methodological weakness of the studies gathered, and mainly due to the impossibility of producing an effect measure for comparisons between interventions, the certainty assessment using

Grading of Recommendations Assessment, Development and Evaluation (GRADE) [56] was considered inappropriate. For this reason, this review does not intend to support recommendations for ML treatment. Even so, this comprehensive review of literature focusing on the typical American mucosal disease and the pooled estimations of the cure rates observed with different interventions are hypotheses generators for future clinical trials. The scarcity and low quality of studies evaluating treatment alternatives for ML are explained by several factors. As a disease that affects neglected populations in underdeveloped countries, the chronic lack of investment in research is widely recognized [57]. In addition, there is an intrinsic difficulty in identifying new drugs against Leishmania [58]. Finally, ML is a relatively infrequent complication of cutaneous leishmaniasis, accounting for approximately 3–6% of cases with skin lesions [8], preventing the execution of large clinical trials.

Other systematic reviews addressing interventions for different clinical forms of leishmaniasis have already been published [1,10,11]. The first, published 15 years ago (2007), also used a comprehensive inclusion criterion in terms of study design, however, only 8 of the 22 previously included studies [1] were selected, due to our definition of a minimum of 10 patients treated in each intervention arm. In 2018, a new review focusing on combined therapies with pentavalent antimonial derivatives was published, but only two studies addressing ML were included [10]. The most recent review was conducted by Pinart *et al*, 2020 [11], updating a review published in 2009 [59]. Based exclusively on RCTs, eight studies addressing ML were retrieved [11].

Considering that the available evidence has not yet been able to answer the clinical questions surrounding the treatment of ML, this study follows the more recent trend of reframing the classic evidence hierarchy pyramid [60]. For conditions not studied in controlled studies, it is important to look for alternatives to optimally promote the production of knowledge and guide practice. Using a broad and systematic search strategy and focusing on critical analysis, aware of the limitation of grouping intrinsically different studies, this approach intends to provide the best available knowledge to support clinical decisions and identify the most promising interventions to be studied in clinical trials.

Despite the high risk of bias involving the selection of population in the retrospective and noncomparative studies, the profile of the population gathered here, based on the age average reported by authors, reflects the patient typically affected by ML, adult male, between the 5th and 6th decade of life, although, there is a wide age range (from 1 month to 90 years). The low representation of children in this review, while it corresponds to the expected–as a secondary complication of the cutaneous form, also alerts to the diversity of a disease that can involve different clinical variations and consequently require different therapeutic approaches. The large variation in disease duration supports this statement, as well as the diversity of affected mucosal sites and of clinical severity patterns. In general, the degree of confidence in the diagnosis of ML was high, and most studies included patients with parasitological or molecular confirmation. On the other hand, few studies have presented the species of *Leishmania* involved. In addition to the heterogeneity of the disease itself, the studies also present great methodological heterogeneity (variation in study design and risk of bias). The high heterogeneity between the studies was confirmed by the wide confidence intervals and moderate to high $I^2$ index observed for almost all intervention groups. Based on that, we chose to critically analyse the clinical and methodological differences among the studies to explain the discrepancies. This review clearly demonstrates that there is a lack of standardization in the definitions of cure, failure, and relapse, including the assessment criteria itself and the time defined for this evaluation, both compromising data gathering. The existence of an attempt to standardize criteria and outcomes in CL trials [14,55], although representing a useful model, minimally this should be validated for use in ML and this first compilation allows scaling current practice in relation

to these definitions. The interventions evaluated in RCTs were MA, MF, SSG, AS, and associations of SSG with allopurinol and of MA with pentoxifylline; however, most studies presented some bias risk. The only low-risk RCT study evaluated the association of MA with pentoxifylline [34] in a limited number of participants (23 patients in total).

For all antimonial therapeutic schemes evaluated (Fig 2), only moderate cure rates were observed, varying from 52.6% (CI: 37.9–66.8%) to 77.4% (CI: 51.4–91.7%). As a factor affecting all interventions evaluated, losses observed during follow-up were considered therapeutic failures, which may have underestimated the cure rate estimation. The pooled cure rates reported by studies using low or standard MA doses were quite similar. It is important to note that the two studies addressing low MA doses [38,39] were small and noncomparative studies with evident bias in the outcome assessment. Regarding the association of MA with pentoxifylline, the findings suggest some benefit on the cure rate. Despite the small sample size of the only trial addressing this association [34], the observations provided by another uncontrolled but larger study reinforce this effect [40]. On the other hand, the cure rates using other combinations (MA plus allopurinol [42] or sulfa [26]) were similar to those observed with MA monotherapy in standard doses. It is important to note that the study describing the highest cure rate with the combination of MA plus sulfa [26] represents a retrospective description of cases with significant bias in all domains, selection, treatment, and outcome assessment. In turn, the apparent lower effectiveness of SSG in relation to MA reinforces that observed in other studies [61] and may suggest that meglumine is the most suitable pentavalent derivative for the Leishmania species prevalent in the Americas. Although it was based on noncontrolled studies, difference can be observed in the cure rate between patients treated with lipid (Fig 3) and deoxycholate amphotericin B formulations (p = 0.0001) (Fig 4). However, the analysis of the studies describing the use of amphotericin deoxycholate clearly indicates that most treatment failures were related to patients who were not able to complete the treatment due to AEs.

In turn, the pooled cure rate grouping the patients treated with pentamidine (Fig 5) represents the experience in only one Brazilian centre. Although cases were reported in two different retrospective studies [28,33], there is some overlap in the observation period, allowing the assumption that some patients were reported more than once, which may have overestimated the cure rate. On the other hand, confidence in the estimated cure rate of miltefosine is higher since the rates observed in randomized and uncontrolled studies were similar.

In addition, the only direct meta-analysis involving two RCTs and comparing MA and MF showed no difference between the interventions. Furthermore, a higher cure rate in four years (80%) compared to D180 (55%) was reported by Sampaio (2019), one of the RCTs addressing miltefosine. Even so, to keep alignment with the cure assessment criterion defined previously, the rate used in the grouping of patients was that observed at D180, which may have underestimated the miltefosine's efficacy in long-term evaluations.

Among the studies that evaluated the treatment based on imidazole drugs, high methodological fragility and variation in the cure rate were observed (Fig 7). The study reporting the highest cure rate [26] consists of a retrospective report of the individual experience in a single service, without details of the diagnostic and cure criteria, classified as very poor-quality evidence.

Finally, the experience with aminosidine consists of two small studies [50,51], both prospective and one of them controlled. The cure rate was uniformly low (Fig 8), less than 50%, suggesting the ineffectiveness of this intervention.

The safety profile for the therapies for ML presented in this review needs to be evaluated with caution, considering the lack of systematic and harmonized safety monitoring procedures in the primary studies. Thus, some level of underreporting is expected [62], making it difficult to estimate the true frequency of occurrence. In addition, the absence of a complete analysis of

the AEs, including seriousness, severity, intensity, causality and expectation limits the toxicity impact assessment. It is important to note that the lack of homogeneity between studies in the definition of AE significantly affects the comparison of the event rate between interventions. Furthermore, differences in the management of the follow-up losses and treatment interruptions due to AE, possibly impacted the observed cure rates. In general, for all interventions, the AEs identified reflect the known toxicity pattern already informed in the manufacturers' technical information and literature [63]. The suspension of treatment due to the occurrence of an AE can be understood as the severity criterion adopted by some authors, but the result evaluated (with only 3.6% of serious AE or that caused the interruption) does not reflect the known concern with the toxicity of therapies for leishmaniasis [6].

The present review has several limitations. However, one of its main contributions was to gather, for the first time, in a systematic and exhaustive way, all descriptions of the occurrence of adverse events available among therapeutic studies addressing ML. Thus, in addition to effectiveness, safety profiles can also be used to support recommendations for the management of ML. For research and public polices, our observations confirm the urgent need for harmonization of clinical outcomes in leishmaniasis. Additionally, the poor, non-systematized and incomplete description of adverse events in terms of frequency, intensity, seriousness and causality also confirms the lack of reliable pharmacovigilance data. These points represent weaknesses to be faced also by health surveillance systems, in order to improve the usefulness of the compulsory notification data collected in endemic countries.

Finally, this review brings important reflections on the treatment of ML. First, there is a lack of quality evidence supporting clinical decisions. Based on the experience gathered, we confirmed an overall only moderate cure rate and a large discrepancy in the published rates, mainly reflecting the lack of standardization in the outcome evaluation criteria but also possibly the existence of prognostic factors related to the different clinical forms, characteristics of patients, and possibly to the parasite, which should be the subject of future studies.

The only possible direct comparison does not suggest the presence of a difference between the interventions, confirming what was observed in the indirect comparison between MA and miltefosine using pooled cure rates. This observation reinforces those biases related to small samples, and uncontrolled studies should be assumed to be present in the effect estimates presented. Thus, the main contribution of this review is to provide a comprehensive overview of the clinical experience in the treatment of ML in the Americas and to point out interventions and possible combinations that are eligible to be compared in a large pragmatic clinical trial.

## Supporting information

**S1 Fig. Risk of bias assessment by study.**
(TIF)

**S1 Table. Preferred Reporting Items for Systematic Reviews and Meta-Analyses check list.**
(DOCX)

**S2 Table. Search strategies performed on December 15th, 2021.**
(DOCX)

**S3 Table. Adverse events according to therapy arm in each study. &**: AE were considered per treatment and not per patient. *the number of participants reporting AE is different from the number of participants reporting a cure rate. **ABLC**: Amphotericin B, lipid complex. **AE:** Adverse Event. **AS**: Aminosidine sulphate. **c-AMB**: Amphotericin B colloidal dispersion. **CTCAE**: Common Terminology Criteria for Adverse Events. **DAIDS**: Division of AIDS Table for Grading of Severity of Adult and Pediatric Adverse Events. **d-AMB**: Deoxycholate

amphotericin B. **HLGT**: High Level Group Terms. **L-AMB**: Liposomal amphotericin B. **MA**: Meglumine antimonate. **MA-LD**: Meglumine antimonate low dose. **MF**: Miltefosine. **NEC**: not elsewhere classified. **NR**: not reported. **PENT**: pentamidine. **PT**: Preferred Term. **Sb$^v$**: Antimonial pentavalent. **SOC**: System Organ Classes. **SSG**: Sodium stibogluconate. (DOCX)

**S4 Table. High Level Group Terms for the Investigation System Organ Classes Term by Therapy. AE:** Adverse Event. **HLGT**: High Level Group Terms. **L-AMB**: Liposomal amphotericin B. **MF**: Miltefosine. **Sb$^v$**: Antimonial pentavalent. **SOC**: System Organ Classes. **SSG**: Sodium stibogluconate. (DOCX)

## Author Contributions

**Conceptualization:** Janaína de Pina Carvalho, Carolina Senra Alves de Souza, Gláucia Cota.

**Data curation:** Janaína de Pina Carvalho, Gláucia Cota.

**Formal analysis:** Janaína de Pina Carvalho, Sarah Nascimento Silva, Mariana Lourenço Freire, Líndicy Leidicy Alves, Carolina Senra Alves de Souza, Gláucia Cota.

**Investigation:** Mariana Lourenço Freire.

**Methodology:** Janaína de Pina Carvalho, Sarah Nascimento Silva, Mariana Lourenço Freire, Líndicy Leidicy Alves, Carolina Senra Alves de Souza, Gláucia Cota.

**Project administration:** Janaína de Pina Carvalho, Sarah Nascimento Silva.

**Software:** Gláucia Cota.

**Writing – original draft:** Janaína de Pina Carvalho, Sarah Nascimento Silva, Mariana Lourenço Freire, Gláucia Cota.

**Writing – review & editing:** Janaína de Pina Carvalho, Sarah Nascimento Silva, Mariana Lourenço Freire, Líndicy Leidicy Alves, Carolina Senra Alves de Souza, Gláucia Cota.

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
