## [Decision Letter · Decision Letter 0]

10 Oct 2022

Dear Mrs Carvalho,

Thank you very much for submitting your manuscript "The cure rate after different treatments for mucosal leishmaniasis in the Americas: a systematic review" for consideration at PLOS Neglected Tropical Diseases. As with all papers reviewed by the journal, your manuscript was reviewed by members of the editorial board and by several independent reviewers. The reviewers appreciated the attention to an important topic. Based on the reviews, we are likely to accept this manuscript for publication, providing that you modify the manuscript according to the review recommendations. 

Sincerely,

Gregory Deye

Academic Editor

Ricardo Fujiwara

Section Editor

Reviewer's Responses to Questions

**Key Review Criteria Required for Acceptance?**

**Methods**

-Are the objectives of the study clearly articulated with a clear testable hypothesis stated?

-Is the study design appropriate to address the stated objectives?

-Is the population clearly described and appropriate for the hypothesis being tested?

-Is the sample size sufficient to ensure adequate power to address the hypothesis being tested?

-Were correct statistical analysis used to support conclusions?

-Are there concerns about ethical or regulatory requirements being met?

Reviewer #1: The authors carried out a systematic review on first and second choice treatments and other diverse treatments of mucosal leishmaniasis (ML) in the Americas with the main objective of updating the evidence on the level of cure of the disease and, as a secondary objective, adverse effects.

The objectives proposed by the authors are perfectly articulated with the hypothesis or question that motivated them.

The systematic review was conducted in different databases and performed by different and independent reviewers. Inclusion and exclusion criteria were established. Risks of bias in the studies were verified. The study was registered in the International Prospective Register of Systematic Reviews (PROSPERO) and followed the guidelines of the Preffered Reporting Items for Systematic Reviews (*PRISMA). The method used for the review was innovative, different from what had been done for the disease until then, as it included studies with different designs (randomized and observational studies, I think this initiative is good idea). I made considerations placed in “corrections”, regarding the cure criterion, Table 2 and in the evaluation by PRISMA.

Regarding the main results, 1104 studies were identified, of which 27 original studies were selected with 10 or more patients (one of the inclusion criteria) and 1666 patients treated for SCI were included.

The sample size was not quite desirable, in order to allow conclusions that would allow generalizations, but this was justified by the scarcity of scientific works in relation to this neglected disease. But, in any case, it was enough to achieve results that confirm results already found in other reviews, for example that antimony is still the first drug of choice used for ML, showing a moderate scientific level of cure, with a possible increase in effectiveness when combined with pentoxifylline) and also new results previously suggested as necessary, in previous studies (miltefosine with a cure level similar to antimony, now under direct evaluation in randomized trials) high toxicity of amphotericin B deoxycholate with low cure level due to high levels of interruptions of the treatment.

The authors comment that this approach intends to provide the best available knowledge to support clinical decisions and identify the most promising interventions to be studied in clinical trials, and I think they do.

As for the statistical analysis: it was used comprehensive Meta-Analysis software v.3.0® to perform a one-group meta-analysis of study arms using a given treatment (pooled rates) based on the latest cure assessment reported in the original studies. Clinical cure rates were calculated according to the intention-to-treat analysis: the analysis was based on the total number of randomly assigned participants, irrespective of how the original study’s authors analyzed the data. These unadjusted indirect comparisons were compared with direct comparisons, when available. They used the inconsistency (I2) statistic to evaluate heterogeneity. Finally, we believe that the statistical analysis used was able to give credibility to the results achieved.

As for approval by the Ethics Committee, I think the project does not directly involve patients because data from articles already published in the literature were used.

Reviewer #2: This is a very well-designed systematic review on mucosal leishmaniasis treatment. Authors have dealt with the peculiarities and the scarcity of published evidence and they have chosen the appropriate methods including those for RCTs and the specific ones for the observational studies. Data extraction and analysis were done in a clear manner and the methods were applied in a correct way considering the intrinsic limitations of the available studies. The systematic review was registered in PROSPERO and followed all the current recommendations for conducting this type of study.

**Results**

-Does the analysis presented match the analysis plan?

-Are the results clearly and completely presented?

-Are the figures (Tables, Images) of sufficient quality for clarity?

Reviewer #1: - Yes, they are in accordance with the analysis plan. The intent-to-treat assessment approach was used. The evaluation period followed Olliaro's criteria after having made an adaptation to the pre-established cure periods. These were presented in percentages with the respective confidence levels. As for the adverse effects (AES), they were presented following the classification in the form of incidence for each type of treatment. I suggested inclusion of a reference for the AES.

The most interesting result was the cure obtained at some point after treatment (D90, 180, D360, values assumed following predetermined cure criteria in the literature). To assess cure, the intention-to-treat method was used, where losses are considered treatment failures. Adverse effects were taken from each study and expressed as percentages with their respective 95% confidence interval levels.

1,104 studies were identified in the databases studied. All exclusion criteria are listed in Fig. 1. Of these 142 studies were read and 27 included (7 randomized and 15 observational -12 retrospective and 3 prospective) 5 non-randomized experimental, and presented a table (Table 2) with the characteristics of the included studies, such as year, author, country treatment with dose and duration, case definition, definition of cure, definition of relapse, cure result and patient follow-up time. Still on Tab. 2 data with the characteristics of the patients studied, such as age, gender, duration of symptoms, type of mucosal lesion, clinical classification of the lesion, whether there was a previous cutaneous lesion and the species of leishmania identified.

The authors explained criteria used to reach a consensus of assessment of these characteristics in the studies.

A third table (Table 3) shows the results of cures and relapses according to intention to treat. It presents the data: year and authors, type of treatment and number of patients treated, results in D90, 180, 360 last level of cure and the level of relapse in results. There are 8 figures: Fig. 2: Pooled cure rate of pentavalent antimonials using a mixed effects analysis; Fig 3. Pooled cure rate including all patients treated with lipid formulations of amphotericin B in the reviewed studies using a mixed effect analysis. Fig 4. Pooled cure rate including all patients treated with deoxycholate amphotericin in the reviewed studies. Fig 5. Pooled cure rate including all patients treated for ML with pentamidine in the reviewed studies. Fig 6. Pooled cure rate including all patients treated for ML with miltefosine in the reviewed studies. Fig 7. Pooled cure rate including all patients treated for ML with imidazole drugs in the reviewed studies using a mixed effect analysis. Fig 8. Pooled cure rate including all patients treated for ML with aminosidine in the reviewed studies. Fig 9. Meta-analysis of studies directly comparing meglumine antimoniate and miltefosine for the treatment of ML.

There is a fourth table on adverse effects with their incidence percentages according to the drug used (Table 4. System Organ Classes for adverse events according to therapy).

When discussing adverse effects, the authors limited themselves to referring to "General disorders and administration site conditions" (such as fever, chest pain, chills, malaise, etc.). I agree, but I suggested that literature studies on AES be remembered.

Then there is a figure (Fig 10. Risk of bias assessment) that shows the outcome of the risk of bias assessment of all included studies.

After these findings, I think that the results are presented in a clear way and that the data necessary for the conclusions are expressed.

Evaluating the tables and figures, these are sufficiently clear. We also suggest in “corrections”, some considerations.

Reviewer #2: The results have followed the PRISMA recommendations and all the essential components were clearly demonstrated. Also the supplementary files were useful to fully understand some of the statements included in the result section.

**Conclusions**

-Are the conclusions supported by the data presented?

-Are the limitations of analysis clearly described?

-Do the authors discuss how these data can be helpful to advance our understanding of the topic under study?

-Is public health relevance addressed?

Reviewer #1: - Yes, the conclusions respond to the objectives and are supported by the data presented.

- Yes, the authors clearly describe limitations such as the number of articles on the subject, limited generalization of conclusions due to the lack of application of GRADE due to the inclusion of non-randomized articles

-.Yes. The authors make this discussion showing the importance of the study for the continuity of other clinical trials using drugs identified as more promising, and to support the ML treatment recommendations. It also shows the tendency to resignify the classic hierarchical pyramid of scientific evidence.

- Public health relevance could be more reinforced.

Reviewer #2: The conclusion is supported by the data under analysis and a very careful list of limitations with their respective explanations were included. In fact, the paragraphs dedicated to describe the study limitations can be very useful for the potential readers not familiarized with the peculiarities of mucocutaneous leishmaniasis in the Americas. Although authors mention something for the next steps in the field of treatment of ML, I feel that more emphasis should be dedicated to the possibility of improving the clinical and therapeutic data registries in the endemic countries in order to have the routine surveillance systems as a source of useful information regarding the challenge of achieving effective treatments. Also, it could be highlighted the urgent need to turn viable the conduction of an adequate powered RCT for comparing the current therapeutic options in a large pragmatic clinical trial involving all the ML cases treated in the Americas.

**Editorial and Data Presentation Modifications?**

Reviewer #1: Modifications to be evaluated:

1- The cure criteria used by the clinical trials are clear in Table 2, but I think they should be clearly expressed in the text, including noting that an improvement of more than 90% was included as a cure. Perhaps this should be addressed in Methods item Outcomes line 144.

2- Lines 536-8: “in general, for all interventions, the AEs identified reflect the known toxicity pattern already informed in the manufacturers' technical information and literature”. I think that a review article on adverse effects of drugs used in the Americas (doi: 10.1016/j.actatropica.2011.02.007) line 538 should be cited here.

3- Lines 71-72: [3,4] have also been implicated with mucosal involvement, in addition to L. infantum and L. major [5]…Here I think it could also be added to L.amazonensis.

4- In item 20 of the checklist of items to include when reporting a systematic review or meta-analysis of PRISMA, the realization of the forest plot is considered ideal. This was not done, we asked why and thought it was worth commenting on it.

5- Figures 2, 3,4,5,6, 7,8,9 regarding cure rates show vertical lines, mostly with negative values (for exemple: -1.00; -0.50, etc.). I wonder if these lines would not be dispensable and if negative values for cure rates would not be inappropriately used (lines 323 -366).

6- In Figure 10 (a and b) in Outcome or Casuality I think it would be better or more appropriate to put moderate risk instead of mean risk.

 7- I had difficulty understanding that the numbers placed in the first cell of Tables 1 and 2 referred to the bibliographic references of the authors, I think it is appropriate to include the word References next to the item ( Year, Author). I think it would also be appropriate to include the total number of patients in the same table to make it clearer.

Reviewer #2: Please see the suggestion in the "Conclusions" comments.

**Summary and General Comments**

Reviewer #1: It helps to fill a gap in ML scientific knowledge to date. It helps in clinical practice and shows, points to results that can help in the study of more effective therapeutic schemes for ML. Public health gains in knowledge regarding leishmaniasis and helps to suppress a lack of knowledge regarding the specific problem of the mucous form, one of the most serious form of the disease . The weaknesses observed in relation to the project relate to the lack of studies and standardization of study methodologies.

Reviewer #2: Please see above.

PLOS authors have the option to publish the peer review history of their article (what does this mean?). If published, this will include your full peer review and any attached files.

Reviewer #1: No

Reviewer #2: Yes: Gustavo Adolfo Sierra Romero

Figure Files:

Data Requirements:

Reproducibility:

References

---

## [Editor Report · Decision Letter 1]

2 Nov 2022

Dear Mrs Carvalho,

We are pleased to inform you that your manuscript 'The cure rate after different treatments for mucosal leishmaniasis in the Americas: a systematic review' has been provisionally accepted for publication in PLOS Neglected Tropical Diseases.

Best regards,

Gregory Deye

Academic Editor

Ricardo Fujiwara

Section Editor

---

## [Editor Report · Acceptance letter]

9 Nov 2022

Dear Mrs Carvalho,

We are delighted to inform you that your manuscript, "The cure rate after different treatments for mucosal leishmaniasis in the Americas: a systematic review," has been formally accepted for publication in PLOS Neglected Tropical Diseases.

Best regards,

Shaden Kamhawi

co-Editor-in-Chief

Paul Brindley

co-Editor-in-Chief
